# Consistent long-term observations of surface phytoplankton functional types from space

Hongyan Xi[1*,] Marine Bretagnon[2], Ehsan Mehdipour[1,3], Julien Demaria[2], Antoine Mangin[2], Astrid Bracher[1,4]

[1]Alfred Wegener Institute, Helmholtz-Centre for Polar and Marine Research, Bremerhaven, 27570, Germany
[2]ACRI-ST, Sophia Antipolis Cedex, France
[3]School of Business, Social & Decision Sciences, Constructor University, Bremen, Germany
[4]Institute of Environmental Physics, University of Bremen, Bremen, 28359, Germany

*Correspondence to*: Hongyan Xi (hongyan.xi@awi.de)

**Abstract.** Global products of phytoplankton functional types (PFTs) derived from multi-sensor ocean color data provide important long-term biogeochemical quantifications indexed by chlorophyll *a* concentration (Chl*a*) of PFTs including diatoms, haptophytes, prokaryotic phytoplankton, dinoflagellates, and green algae. Due to the distinctive lifespans and radiometric characteristics of ocean color sensors, the consistency of the PFT products derived from different sensors needs to be assured to establish a complete and systematic frame for long-term monitoring of multiple PFTs on a global scale. This study introduces a machine learning (ML) based correction scheme to eliminate the discrepancies between different sensors' PFT products. The correction scheme is applied to the Sentinel 3A/B Ocean and Land Colour Instrument (OLCI) derived PFT data, to match them with the PFT data derived from GlobColour merged ocean color products using the overlapped period. This correction has generated consistent PFT data across the sensors, enabling the analyses of multi-year PFT observations by describing their variability and two-decade trends. Analysis of PFT time series has revealed an increasing trend in diatoms and dinoflagellates and a decreasing trend in haptophytes and prokaryotic phytoplankton on a global scale. The overall trend in green algae remains relatively stable, although with some spatial variations. These PFT trends are more significant in high latitudes and coastal regions, and for prokaryotic phytoplankton also in the equatorial region. Anomaly of PFTs in 2023 shows significant increases in Chl*a* of diatoms and dinoflagellates (+24 % and +9.4 %, respectively), but only weak changes in Chl*a* for prokaryotic phytoplankton (-2.1 %) and haptophytes (~1.6 %). These consistent time series data will act as an important ocean indicator to infer possible changes in the marine environment.

## 1 Introduction

Climate-induced changes stress the ocean's contemporary biogeochemical cycles and ecosystems, impacting the base of the marine food web - phytoplankton communities (Gruber et al., 2021). In the past decades, various observations of ocean color (OC) information, especially the chlorophyll *a* concentration (Chl*a*) as a proxy of phytoplankton biomass, have been able to revolutionize our understanding of the marine biogeochemical processes and provide insights of the changes in phytoplankton (e.g., Antoine et al., 2005; Gregg and Rousseaux, 2014; Behrenfeld et al., 2016). However, phytoplankton biomass cannot comprehensively describe the complex nature of the phytoplankton community, concerning their composition and function. Phytoplankton community composition varies in ocean biomes and phytoplankton groups drive differently the marine ecosystem and biogeochemical processes (Bracher et al., 2017). Therefore, continuous long-term monitoring of phytoplankton functional types (PFTs) with inter-annual variation and trend analysis will help understand better the biogeochemical processes and benefit the assessment of ocean health (Xi et al., 2023a).

Previously, we have developed and further improved an approach, referred to as EOF-PFT, consisting of a set of empirical-orthogonal-function based algorithms for the retrieval of PFTs on a global scale (Xi et al., 2020; 2021). Two algorithms within the EOF-PFT approach were built for two sets of OC satellite products, namely the GlobColour merged products with sensors of SeaWiFS, MODIS-Aqua, MERIS and VIIRS-SNPP included, and the products from the OLCI (Ocean and Land Color

Instrument) sensors onboard Sentinel 3A and 3B. Using multi-spectral remote sensing reflectance data (Rrs) from these OC products and sea surface temperature (SST) data, the EOF-PFT approach enables satellite retrievals of Chl$a$ for five PFTs with pixel-by-pixel uncertainty, which include diatoms, dinoflagellates, haptophytes, green algae and prokaryotic phytoplankton (prokaryotes hereafter for brevity). These PFT Chl$a$ products, covering the period from 2002 until today, are available on Copernicus Marine Service and updated regularly upon reprocessing with refined algorithms.

The PFT products enable the analysis of multi-year PFT observations by describing their variability and trends. However, prior to the time series analysis, the consistency of the PFT datasets derived from the GlobColour merged OC products and that from OLCI data needs to be assured. In the frame of the Copernicus Marine Evolution Project GLOPHYTS, we aim to merge the aforementioned two PFT datasets into one long-term consistent satellite PFT product. A first attempt has been carried out by Xi et al. (2023a) with a correction scheme based on linear regressions with PFT uncertainty considered, which was applied to PFT data from Sentinel 3A/B OLCI sensors to generate PFT time series in the Atlantic Ocean. Though such a straightforward correction scheme provides an overall consistent time series, the spatial variation cannot be adequately corrected and large biases between sensors can still exist at regional scales. Therefore, we intend to enhance the correction procedure by incorporating spatial variability. In this study, we propose a new correction scheme based on a random forest machine learning method for delivering two-decades quality-assured global PFT datasets, that are cross-validated within model training and further validated with in-situ data. The harmonized PFT time series with high spatio-temporal consistency are analysed on both global and regional scales to investigate the trend and anomaly for different PFTs. Considering that ocean color missions are planned to be continued into the next decade and beyond, such PFT time series will further act as an important ocean indicator to help sustain the ocean health by providing inter-annual variation and trend analyses of the surface phytoplankton community composition, especially for the key regions that have been defined as vital marine environments by the Copernicus Marine Service.

## 2 Data and Methodology

### 2.1 PFT products from Copernicus Marine Service

The PFT datasets with per-pixel uncertainty (product ref. no. 1 in Table 1) are produced with a modified version of the EOF-PFT approach proposed by Xi et al. (2021). The modified algorithms within EOF-PFT were developed using the latest global in situ pigment matchup dataset and trained separately for the merged OC products (including SeaWiFS, MODIS, MERIS and VIIRS) since 2002 with 8 bands and Sentinel 3A/B OLCI data (since May 2016) with 10 bands from GlobColour archive. The official PFT data (product ref. no. 1 in Table 1) are generated from the merged OC products for the period of July 2002–April 2016, and from OLCI products from May 2016 onwards (hereafter referred to as merged sensor-derived PFTs and OLCI-derived PFTs, respectively). However, we extended the merged sensor-derived PFT products to April 2017 in this study (product ref. no. 2 in Table 1), in order to have the one-year overlapping period with the OLCI-derived PFT data for consistency analysis. The merged sensor-derived PFT products were processed only until 2017 because VIIRS-SNPP data from NASA release R2018 reprocessed version have been identified with significant trends (possibly due to degradation) after 2017 that are not identified in other sensors (NASA Ocean Color, last access in May 2025).

Updated EOF-PFT algorithms have also been assessed with an independent validation dataset with satisfactory performance (details in the corresponding QUID). The corresponding prototypes have been prepared and implemented into the Copernicus Marine Service to generate reprocessed PFT products with per-pixel uncertainty through EiS (Enter into Service) by November 2024. With these updates, we obtained PFT retrievals from the aforementioned two sensor sets, however, consistency between the PFT data across the two sets must be assured to generate long-term time series data and prepare for the next generation reprocessing.

**Table 1: Products used.**

| Product ref. no | Product ID and type | Data access | Documentation |
|---|---|---|---|
| 1 | OCEANCOLOUR_GLO_BGC_L3_MY_009_103; satellite observations | EU Copernicus Marine Service Product (2024) | Quality Information Document (QUID): Jutard et al. (2024); Product User Manual: Colella et al. (2024) |
| 2 | Self-processed PFT data based on merged OC products for the period of May 2016–April 2017 overlapped with the OLCI based PFT data available on the Copernicus Marine Service; satellite observations | Our own archive | Xi et al. (2021, 2023a) |
| 3 | In situ PFT data; in situ observations | Our own archive | PANGAEA (Xi et al., 2025) |

## 2.2 Machine Learning Based Ensemble (MLBE) for inter-sensor correction of PFT data

The merged sensor-derived PFT products have a longer timespan (~ 15 years) than the OLCI (Sentinel-3A/B) derived PFTs (~ 7 years) and are generated based on the algorithms trained with a larger global matchup dataset (~ 1500 data points compared to ~ 300 for OLCI due to its shorter running time and limited in situ data from 2016). The merged sensor-derived products also carry relatively lower uncertainty compared to the OLCI-derived PFT data (Xi et al., 2021; 2023a). Therefore, we set up the modification scheme for the OLCI-derived PFTs to match the merged sensor-derived PFTs. A similar inter-sensor correction has been done for the OC-CCI merged OC data (Mélin and Franz, 2014; Sathyendranath et al., 2019). We tested a few machine learning methods (random forest, 1D-convolutional neural network, self-organizing map) to upgrade the OLCI-derived products consistency with the merged sensor-derived products on a pixel basis. At last, we used the random forest-based ensemble 'TreeBagger' with regression decision trees embedded in MATLAB (R2023b) which selects a subset of predictors for each decision split by the random forest algorithm to establish the correction model (Breiman, 2001). The ensemble is powerful in extracting spatial features from the predictors and establishing connections with the response variables through an optimal number of regression trees. Figure 1a shows a simplified flowchart of this machine learning ensemble, which is referred to as MLBE (Machine Learning Based Ensemble) hereafter. A brief description of the ensemble establishment is as follows.

1) Input data are the monthly PFT products with 25-km resolution derived from both merged sensor and OLCI data (May 2016 to April 2017, product ref. no. 2 in Table 1), from which the latitude, longitude, and OLCI-derived PFT products during the 12 months are the predictor variables, and the merged sensor-derived PFTs are response data. Only pixels with available data from both products were taken into account. The input dataset was randomly divided into training (70 %, ~3 million pixels) and testing dataset (30 %, ~1.26 million pixels). Before performing the training, the PFT data sets were log-transformed due to their nature of log-normal distribution (Xi et al. 2021). The geographic information (latitude and longitude) was simply normalized to the range [-1,1] by scaling their original ranges of [-89.875, 89.875] and [-179.875, 179.875] (with 0.25° pixel size).

2) The MLBE was trained separately for each PFT. After testing different numbers of regression trees for the training, we chose 30 regression trees to get the optimal training performance with relatively low computation cost (Figure 1b). Trained models applied to the test datasets have shown equivalent performance with the training sets, indicating that the ensembles are robust.

3) The ensembles trained for the five PFTs (diatoms, haptophytes, dinoflagellates, prokaryotes and green algae) were applied to all monthly PFT products derived from OLCI from May 2016 to December 2023 to generate the corrected OLCI PFT data.

Following the same steps above, a similar MBLE model based on the PFT products with 4-km spatial resolution has also been established to enable the validation with in situ data as described below in Section 2.3, as the corrected OLCI PFT generated from the 25-km based MLBE model is too coarse to have a valid comparison with the field measurements. PFT time series analysis is however still based on the monthly 25-km product to alleviate the computation.

## 2.3 Validation data

We compiled two in situ PFT datasets to validate the MBLE-corrected OLCI-derived PFT products (product ref. no. 3 in Table 1). The in situ data were derived from quality-controlled in situ HPLC pigment concentrations using the diagnostic pigment analysis (DPA) with updated pigment-specific weighting coefficients following Xi et al. (2023a; 2023b), consistent with the calculation of the in situ PFT data used for the updated EOF-PFT algorithms described in Section 2.1. Dataset 1 is the test dataset (99 matchups) extracted from the global in situ PFT matchup data, which takes up 30 % of the whole matchup dataset, while the other 70 % was used for the retuning of the PFT algorithm for OLCI sensors. Dataset 1 spans from 2016 to 2021 and spreads widely in the global ocean. Dataset 2 containing 134 matchups is a newly compiled dataset that composites in situ PFT data collected from four recent mostly polar expeditions with the research vessel *Polarstern* (Alfred-Wegener-Institut Helmholtz-Zentrum für Polar- und Meeresforschung, 2017), that are PS126 (May–June 2021), PS131/1 (June–Aug 2022) and PS136 (May–June 2023) in the north Atlantic to the Arctic Ocean, and PS133 (Oct–Nov 2022) in the Southern Ocean. Geographical distribution maps of the two datasets are included in Figure 3 together with the validation plots. These matchup data are made available upon publication on PANGAEA (in review): https://doi.org/10.1594/PANGAEA.954738.

## 2.4 Trend and anomaly analysis

We focus on explorations of the consistent PFT products to reveal and understand the trends and variations of the global PFTs in the last two decades. We prepared time series on global scale and four regional scales including the North Atlantic Ocean, the Mediterranean Sea, the Arctic Ocean and the Southern Ocean. The other two regions of Copernicus Marine Service's interest, the Baltic Sea and the Black Sea, were not included as the PFT algorithms were developed for open ocean waters (bathymetry > 200 m) and the quality of the PFT data generated in these regions could not be assured (Xi et al., 2021). PFT time series of different spatial scales were calculated by applying the weighted average (taking cosine of the latitude as weights) to the monthly PFT data over the defined regions, to take into account the proportional contribution of each pixel to the global surface ocean due to area distortion in the gridded dataset. The latitude-weighted averaging was applied to the logarithmically transformed PFT Chl*a* to get the log-based mean which are then converted to natural values. A deseasonalization, referring to the process of removing the signal caused by seasonality from the time series, was first applied to the PFT time series. The deseasonalized time series were then prepared by decomposing the monthly data of each variable into a trend, seasonal and residual components with Seasonal-Trend decomposition using Loess (STL: Cleveland et al.,1990). Non-parametric Mann-Kendall test was used to identify statistically significant trends over time with p-value <0.05 (Mann 1945, Kendall 1975, Gilbert 1987), and then the slope of the linear trend was estimated with the non-parametric Sen's slope (Sen, 1968). The standard deviation of the trend slope has been also calculated by considering PFT uncertainty assessed by the EOF-PFT retrieval algorithms. Time series analysis has been done both, per-pixel and for the whole global ocean and selected regions. We detected trends reflected by the satellite observations and derived anomalies to observe the inter-annual changes. Anomalies of 2023 (the last year of the considered period) were also obtained following Xi et al. (2023a) by comparing the PFT situation of 2023 to the mean of the last two decades.

**2.5 Statistical metrics**

To evaluate the correction ensemble performance, relative difference (RD), median absolute difference (MAD) and median absolute relative difference (MARD) have been calculated based on the Chl*a* data of each PFT, which are defined as below. RD$_i$ = (Chl$a_i^{OLCI}$ - Chl$a_i^{Merged}$) / Chl$a_i^{Merged}$, where $i$ is the $i$th PFT

$$RD_{PFT} = \frac{(Chla\_PFT_{OLCI} - Chla\_PFT_{merged})}{Chla\_PFT_{merged}} * 100\% \qquad (1)$$

$$MAD_{PFT} = \text{median of } \left| Chla_{PFT_{OLCI}} - Chla_{PFT_{merged}} \right| \qquad (2)$$

$$MARD_{PFT} = \text{median of } \frac{\left| Chla\_PFT_{OLCI} - Chla\_PFT_{merged} \right|}{Chla\_PFT_{merged}} * 100\% \qquad (3)$$

To validate the corrected PFT Chl*a* data with in situ data, statistical metrics including regression slope and intercept, determination coefficient ($R^2$), root mean square difference (RMSD, mg m$^{-3}$), and median percent difference (MDPD, %) have been used. For the definition equations of these terms please refer to Xi et al. (2020). Note that only the slope and $R^2$ are calculated on the base 10 logarithmic scale.

# 3 Results

## 3.1 Correction of the OLCI-derived PFT data using the MLBE scheme

To reduce cross-sensor data shift and generate consistent PFTs, we have first applied a correction method while using the type II regression relationships with uncertainties included between the merged sensor-derived PFTs and OLCI-derived PFTs in the overlapped period, to correct the latter to the former. The methodology was described in Xi et al. (2023a). However, even though the final PFT time series over the global ocean shows good consistency, the difference between the two PFT products is still prominent in different regions. Taking the diatom product as a showcase, we calculated the relative difference (RD in %) between the OLCI-derived and merged sensor-derived diatom Chl*a* using Equation (1). The median absolute relative difference (MARD in %) calculated using Equation (3) over the globe was reduced significantly after the linear correction (from 45 % to 26 %), nevertheless, the RD can still reach as high as 80 %–100 % in different regions (figure not shown). High RD variations have also been found for other PFTs with the previously proposed correction scheme based on type II linear regression.

The scatterplot and statistics in Figure 1d with the MLBE-corrected OLCI diatom Chl*a* show significant improvement in consistency with the merged sensor-derived diatom retrievals, compared to the non-corrected OLCI-derived diatom data (Figure 1c). Figure 1f highlights the reduced RD variation over the global ocean compared to the RD between the non-corrected OLCI and merged sensor-derived PFTs shown in Figure 1e. The slope of the regression when using the corrected dataset is close to one, the median absolute difference (MAD, defined in Equation (2)) reduced from 0.13 mg m$^{-3}$ to 0.02 mg m$^{-3}$, and the MARD from 45 % to 5.7 %. The trained ensembles applied to the other four PFT products (haptophytes, dinoflagellates, prokaryotes, and green algae, see Figure 2 for the global distribution of the RD for each) have also shown significant improvements with MAD of 0.002, 0.002, 0.003, 0.006 mg m$^{-3}$, and MARD of 5.2 %, 4.2 %, 4.8 %, and 7.2 %, respectively. The median of RD over the globe for all five PFTs is within ±1.5 % and shows no significant over-/underestimation.

The low RD observed for the overlapping year suggests that the MLBE correction scheme effectively aligns the OLCI-derived PFTs data with the merged sensor-derived PFTs, ensuring a strong spatial correspondence between the two datasets.

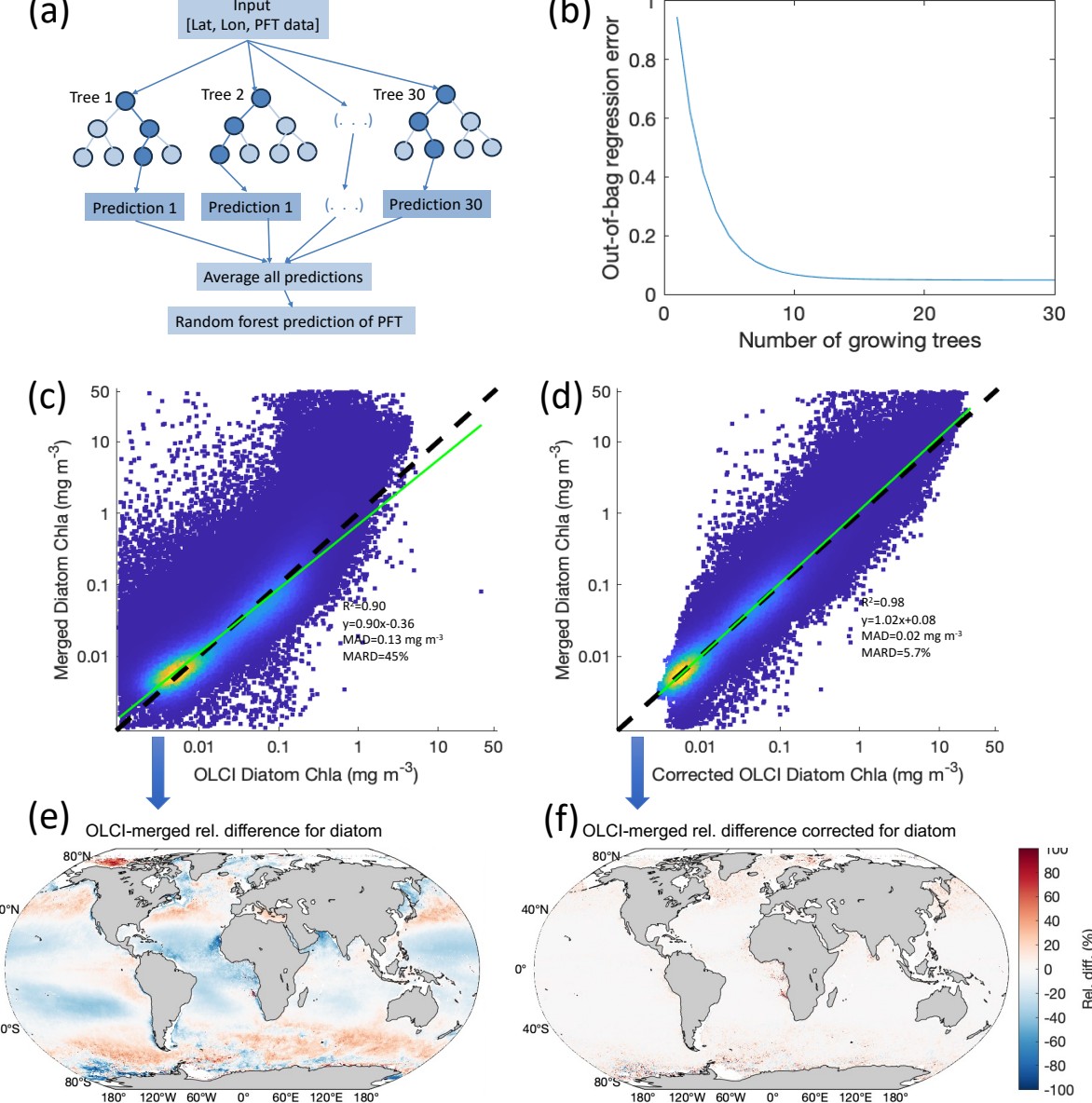

Figure 1: (a) Flowchart of the MLBE; (b) ensemble error with number of growing trees, scatterplots of (c) diatom Chl*a* from OLCI non-corrected against that from merged sensor products and (d) MLBE corrected OLCI diatom Chl*a* against that from merged sensor products, (e) RD between OLCI-based and merged sensor-derived diatom Chl*a*, and (f) RD between MLBE corrected OLCI-based and merged sensor-derived diatom Chl*a*.

185

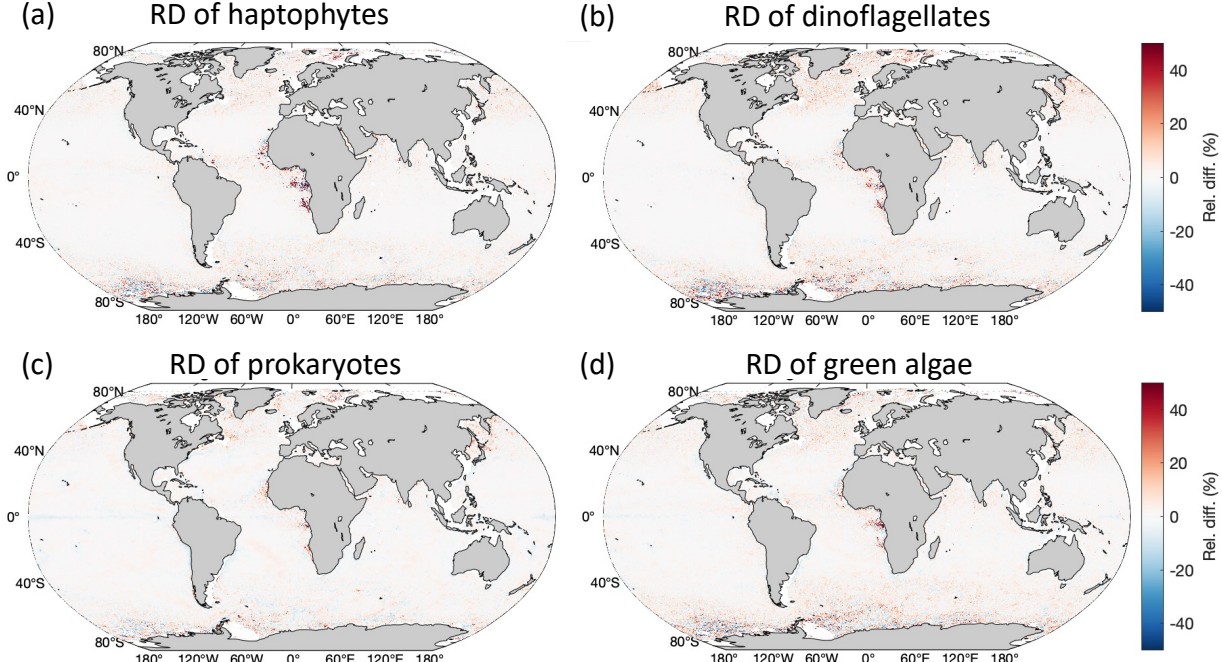

**Figure 2: Global distribution of the RD between MLBE corrected OLCI and merged sensor-derived PFT Ch*la* over one-year overlapped period (May 2016–April 2017).**

### 3.2 Validation of the MLBE-corrected OLCI-derived PFT data

Validation of the corrected OLCI-derived PFTs has been carried out by applying the 4-km based MLBE to the OLCI-derived PFT data that are collocated with the two independent in situ data sets as described in Section 2.3. Scatterplots and statistics of the validation using dataset 1 displayed in Figure 3a show good agreements between the corrected OLCI PFT data and the in situ with $R^2 > 0.51$ and MDPD < 56%, with diatoms showing the best slope (0.78) and correlation coefficient (0.80) and prokaryotes the lowest MDPD (33.5%). We also provided a similar validation analysis for the OLCI data before the correction (Figure S1 in the supplementary document) to have a direct comparison. The overall validation shows that the MLBE correction on the OLCI-derived PFT data preserves the distribution features from the original OLCI-derived PFT data set, however, overall slightly downgraded statistics have been observed for nearly all PFTs, except for the MLBE corrected haptophytes and prokaryotes which showed slightly better MDPD against the in situ compared to the validation of the original OLCI-derived data. The validation using data set 1 indicates that the MBLE correction does not significantly change the PFT variability, showing its feasibility to generate consistent time series data. On the other hand, validation using dataset 2, that contains recently obtained in situ data in high latitudes only, has exhibited larger discrepancies than that from dataset 1 (Figure 3b). All PFTs showed low correlation between the MLBE corrected and in situ data with the highest $R^2$ only 0.21 for diatoms and lowest for green algae. Though the MDPD values are all below 60%, the low R2 indicates weak agreements between the corrected and in situ. Prokaryotes show underestimations in the corrected OLCI data compared to the in situ, mostly for the Arctic data. A similar validation for the original OLCI-derived PFTs using data set 2 has also been provided in Figure S2 in the supplementary document, showing overall almost equivalent statistics with the validation of the corrected data, with a slightly higher $R^2$ of 0.24 for diatoms and also the lowest for green algae (0.09). This confirms that PFT data in high latitudes bear large uncertainties, which is in line with the per-pixel uncertainty estimated by considering errors induced by the input satellite data and the EOF-PFT algorithm parameters (Xi et al. 2021). The satellite PFTs have not been improved even with the MLBE correction, suggesting that the inherent high uncertainties in high latitudes are mostly attributed to the retrieval models that are not efficient enough in these regions. Therefore, PFT observations in the high latitudes need more attention in terms of improved estimation methods and higher data quality.

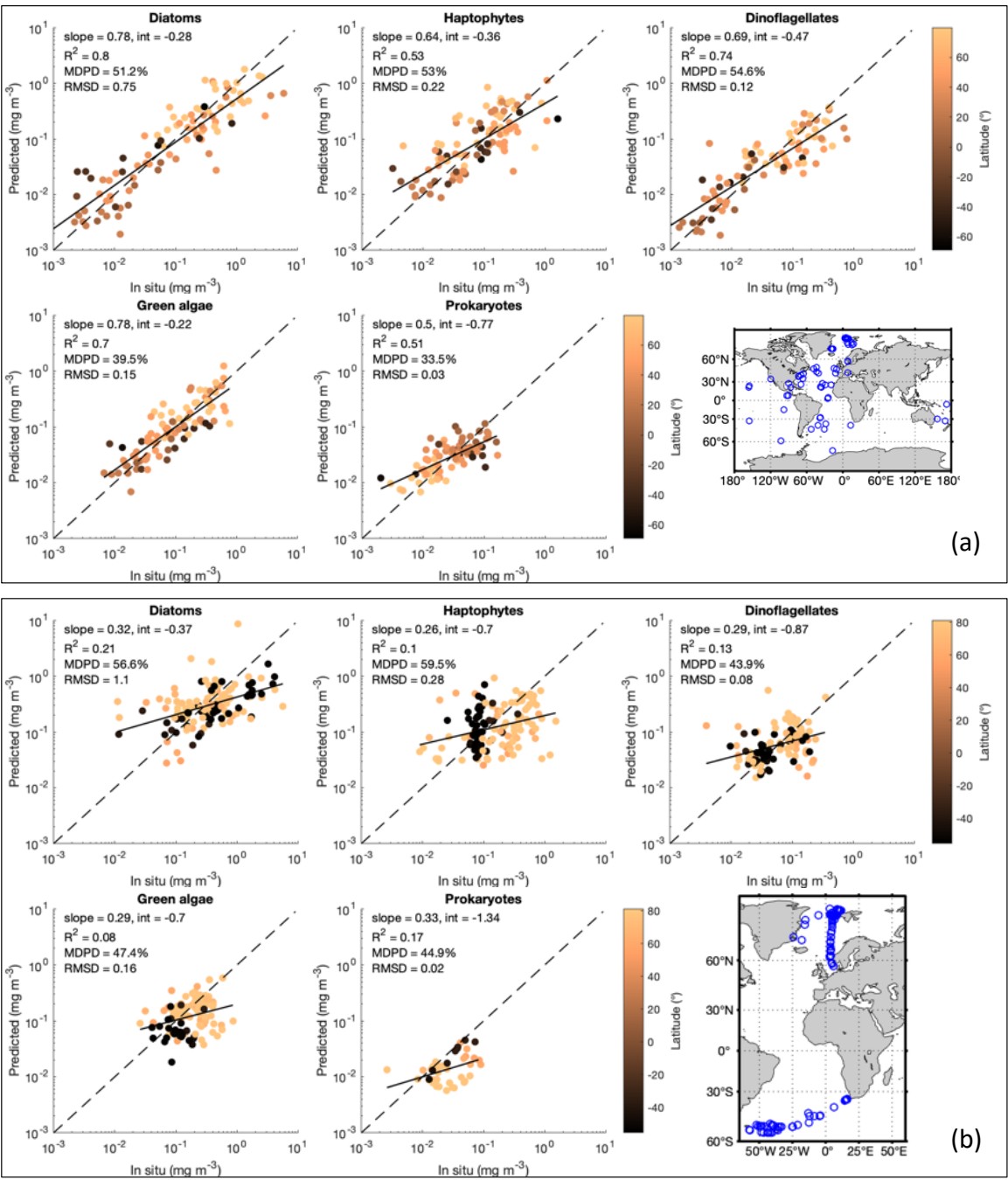

**Figure 3: Panel (a): MLBE-corrected PFT Chl*a* from OLCI sensors in comparison with in situ PFT dataset 1. Panel (b): same as the validation in Panel (a) but using the in situ PFT dataset 2. A map of the data distribution for datasets 1 and 2 (product ref. no. 3 in Table 1) is shown in each panel respectively.**

### 3.3 PFT time series analysis

We applied the MLBE correction scheme on global scale to the OLCI monthly products and generated time series for the five PFTs from July 2002 to December 2023. With the corrections applied to OLCI data, all five PFTs show very consistent time series (Figure 4a). The MLBE corrected OLCI-derived PFT data and the merged sensor-derived PFT data have shown almost identical values during the overlapped period (May 2016–April 2017). Only for green algae the correction is slightly less satisfactory than the others, which should be due to the weaker correlation ($R^2 < 0.7$, figure not shown) between the original OLCI and merged sensor-derived PFT data whereas the other four PFTs show $R^2$ all above 0.9. This weaker correlation for green algae has subsequently led to reduced performance in the MLBE correction. The PFT time series have been analysed at the global scale and four regional scales including the North Atlantic Ocean, the Mediterranean Sea, the Arctic Ocean and the Southern Ocean. Figure 4b shows the time series with slopes indicating the PFT trends per decade and the corresponding slope

errors for all the PFTs at different scales. Figure 5 shows the significant PFT trends (p-value < 0.05) on a pixel basis over the globe to have a better understanding of the spatial distribution of the trends.

Diatoms show a significant increasing trend for the global ocean and selected regions, especially in the Atlantic section of the polar regions (Figure 5a). Further, a distinct increase has been found in more recent years since autumn 2017 and is still prominent in 2023. The global trend of diatom Chl$a$ is increasing by $0.0011 \pm 0.0001$ mg m$^{-3}$ per decade and with a dramatic increase in the polar regions (0.03 and 0.034 mg m$^{-3}$ per decade for the Southern Ocean and the Arctic Ocean, respectively). This overall increasing trend is mainly driven by the significant elevation in diatom biomass observed since 2018, especially due to the higher minimum diatom Chl$a$ in spring and late autumn which are the beginning and ending time of the available OC satellite observations in the polar areas. This might suggest a longer growth period for diatoms in latest years.

Haptophytes Chl$a$ exhibits a very slight decrease in general on the global scale ($-0.0002 \pm 0.0001$ mg m$^{-3}$ per decade) and all other selected regional zones but the decrease is not significant in the North Atlantic Ocean. There is a slight oscillation pattern in the global time series, which shows the haptophyte biomass was the highest in late summer 2011, and remained at a stably lower biomass in the following years until 2018/2019 when it started to elevate again. This feature is not clearly reflected in the selected four regions therefore should be attributed to other regions that are not included here. Global per-pixel trend (Figure 5b) shows a more significant decrease in coastal areas, the sub-Arctic and Arctic regions, and high variability in the Southern Ocean with an overall decrease.

Dinoflagellates show a similar pattern with diatoms, i.e., an increasing trend ($0.0002 \pm 0.0000$ mg m$^{-3}$ per decade) in the last two decades mainly driven by the increase in dinoflagellates Chl$a$ since mid-2017, but their biomass is still low compared to other PFTs as they are usually undominant in the phytoplankton community composition. No significant trends have been found for dinoflagellates biomass in the Mediterranean Sea and the Arctic Ocean.

Green algae show no significant trend on the global scale. The time series show a less obvious seasonal pattern than the other PFTs, possibly due to that they are barely the dominant group in the global ocean and mostly co-exist with the other PFTs which show clear dominance in certain regions at specific times, depending on their ecological functions. The biomass reached its peak in October 2011, followed by a few years of decrease but started to increase in 2018. On the regional scale, a decrease in the Mediterranean Sea and the Arctic Ocean, and a slight increase in the Southern Ocean have been observed, which is also clearly shown in the per-pixel trend (Figure 5d). The decreasing trend is seen in coastal regions such as the north European coastlines, the west coast of America and Africa, and the north coast of the Arabian Sea.

Prokaryotes Chl$a$ displays an overall significant decreasing trend on the global scale ($-0.0012 \pm 0.0001$ mg m$^{-3}$ per decade) and the selected regional zones except for the Southern Ocean. Global per-pixel trend (Figure 5e) shows the north hemisphere with significant decrease near the equator within 15° S–25° N (Indian Ocean, West Africa, low latitudes in the Pacific Ocean), but a slight increase is shown in the belt of 15° S–35° S. Very mild changes have been found in high latitudes where the prokaryotic phytoplankton abundance is in general very low ($\ll 0.01$ mg m$^{-3}$ on area average).

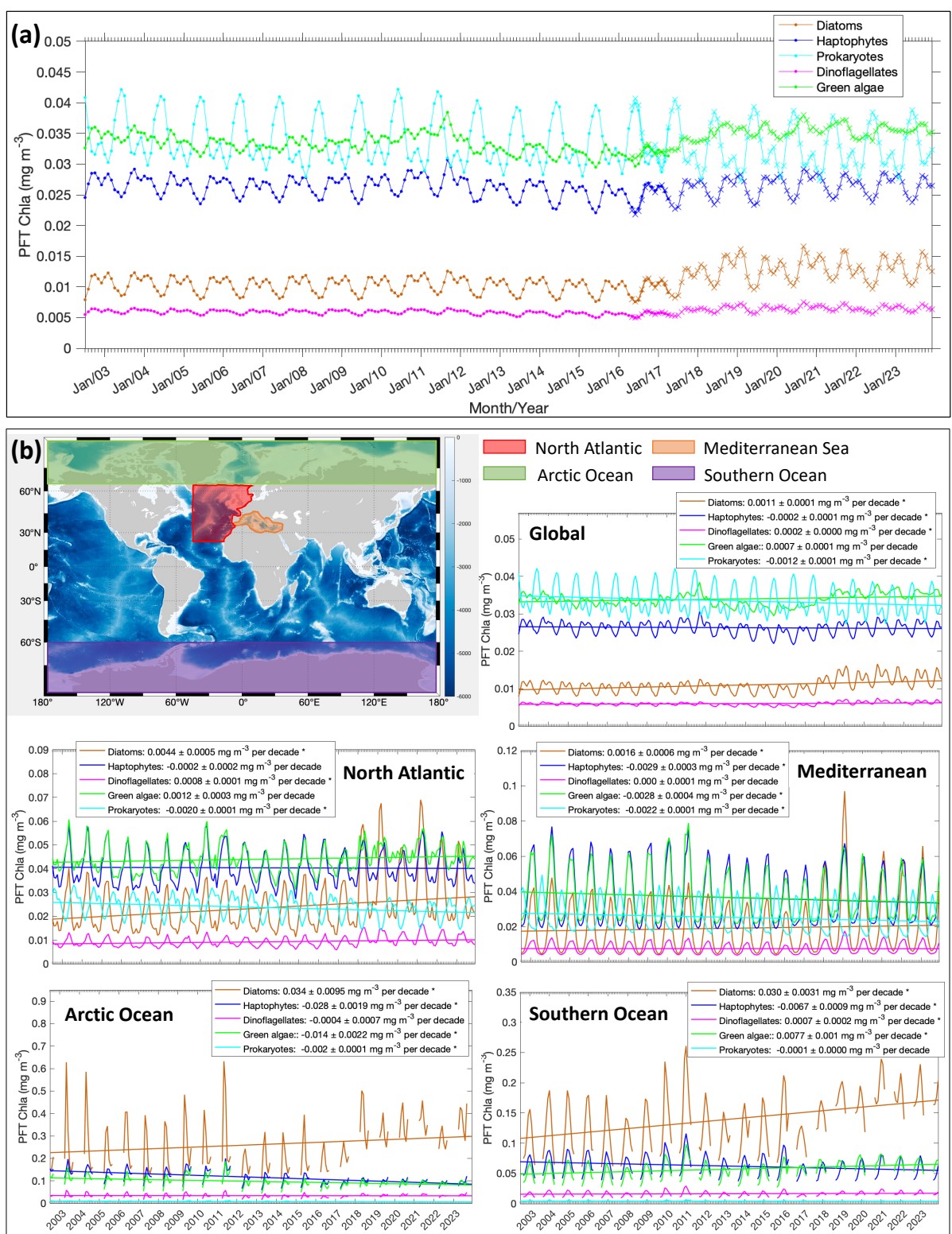

**Figure 4. Panel (a):** Updated (corrected) time series of the five PFT Chl*a* based on the global mean from 2002 to 2023. Merged sensor-derived PFT products cover the period of July 2002-April 2017 (indicated with dots), and OLCI-derived PFT products are for May 2016-Dec 2023 (indicated with crosses). Note that the OLCI-derived products have been corrected to merged products based on MLBE. **Panel (b):** Trends of diatoms, haptophytes, dinoflagellates, green algae and prokaryotes Chl*a* on the global scale and four regional scales (the North Atlantic Ocean, the Mediterranean Sea, the Arctic Ocean and the Southern Ocean), respectively. Trend slopes per decade with uncertainties have been indicated with significant trends marked with an asterisk (*).

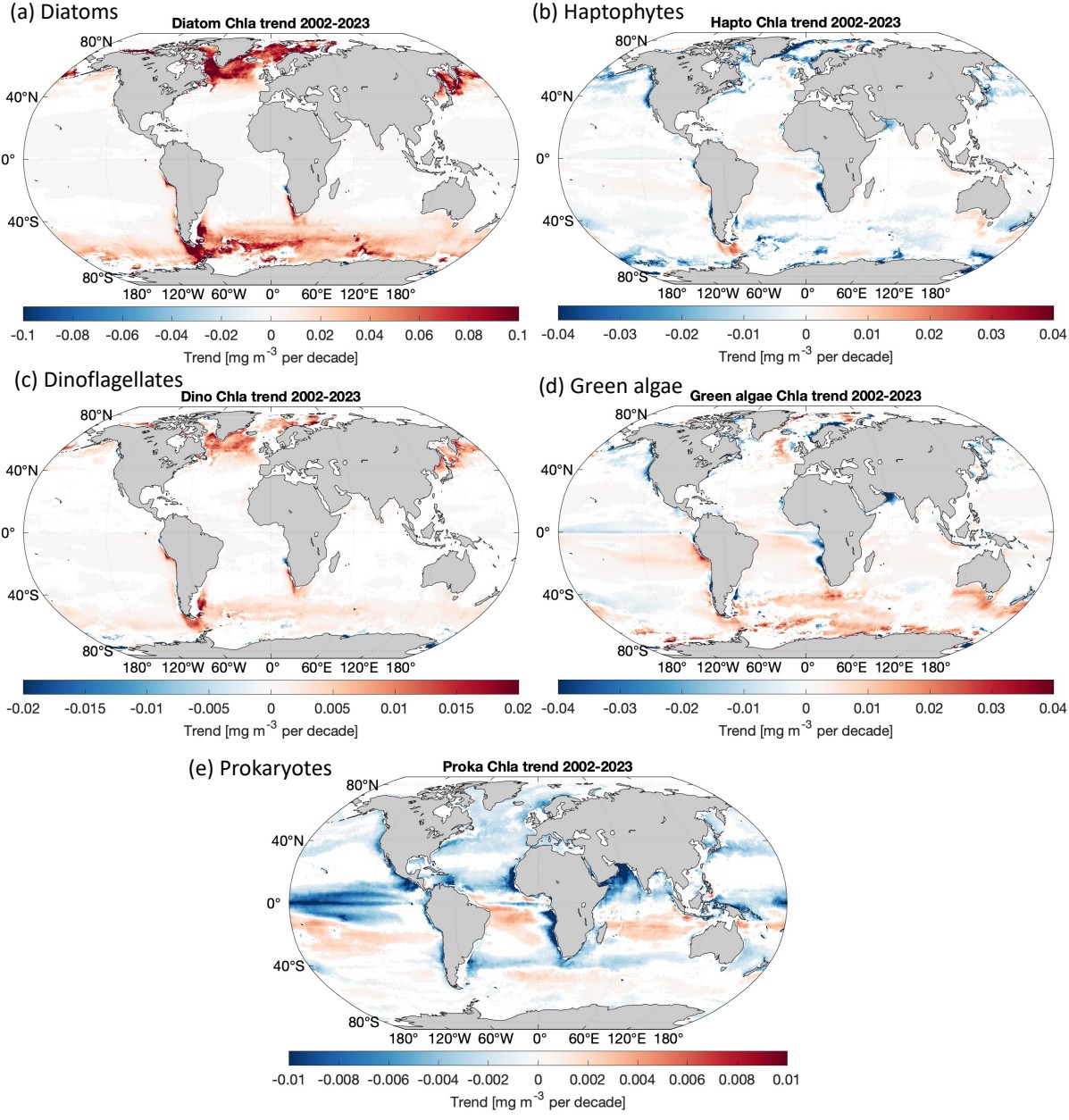

**Figure 5. Per-pixel trends for Chl*a* of (a) diatoms, (b) haptophytes, (c) dinoflagellates, (d) green algae, and (e) prokaryotes (only where p < 0.05 is shown; slope unit: mg m⁻³ per decade).**

### 3.4 PFT anomaly of 2023

Figure 6 shows the relative anomalies (%) of the five PFTs in 2023 compared to the average PFT state over the 20 years. The diatom anomaly presents higher Chl*a* for most of the global ocean with a dramatic increase in latitudes > 40°. This can already be expected from the time series in Figure 4a where diatoms show elevated Chl*a* since autumn 2017 and keep similar higher biomass in 2023. The global mean of the diatom in 2023 is about 24% higher than the two-decade average, and the anomaly varies from -30% to 110% with extremely high values in the Arctic Ocean and the coastal regions in the southern part of South

America. Dinoflagellates show a similar anomaly with diatoms in a much milder pattern, which has a global mean of about 9.4%. The haptophyte anomaly presents changes without a clear pattern, showing slight increases in Chl*a* in the Pacific gyres, eastern Indian Ocean and the Southern Ocean, but slight decreases in the temperate latitudes. The overall global mean anomaly of haptophytes Chl*a* is only very slightly higher compared to the two-decade average (1.6%). Green algae show a similar distribution in biomass change as haptophytes but a bit more prominent increase in most of the global oceans (global mean of

6.5%). Prokaryotes show in general decreased Chl*a* in 2023 (global mean of -2.1%), with only slight increases observed in the South Pacific Ocean and part of the Southern Ocean.

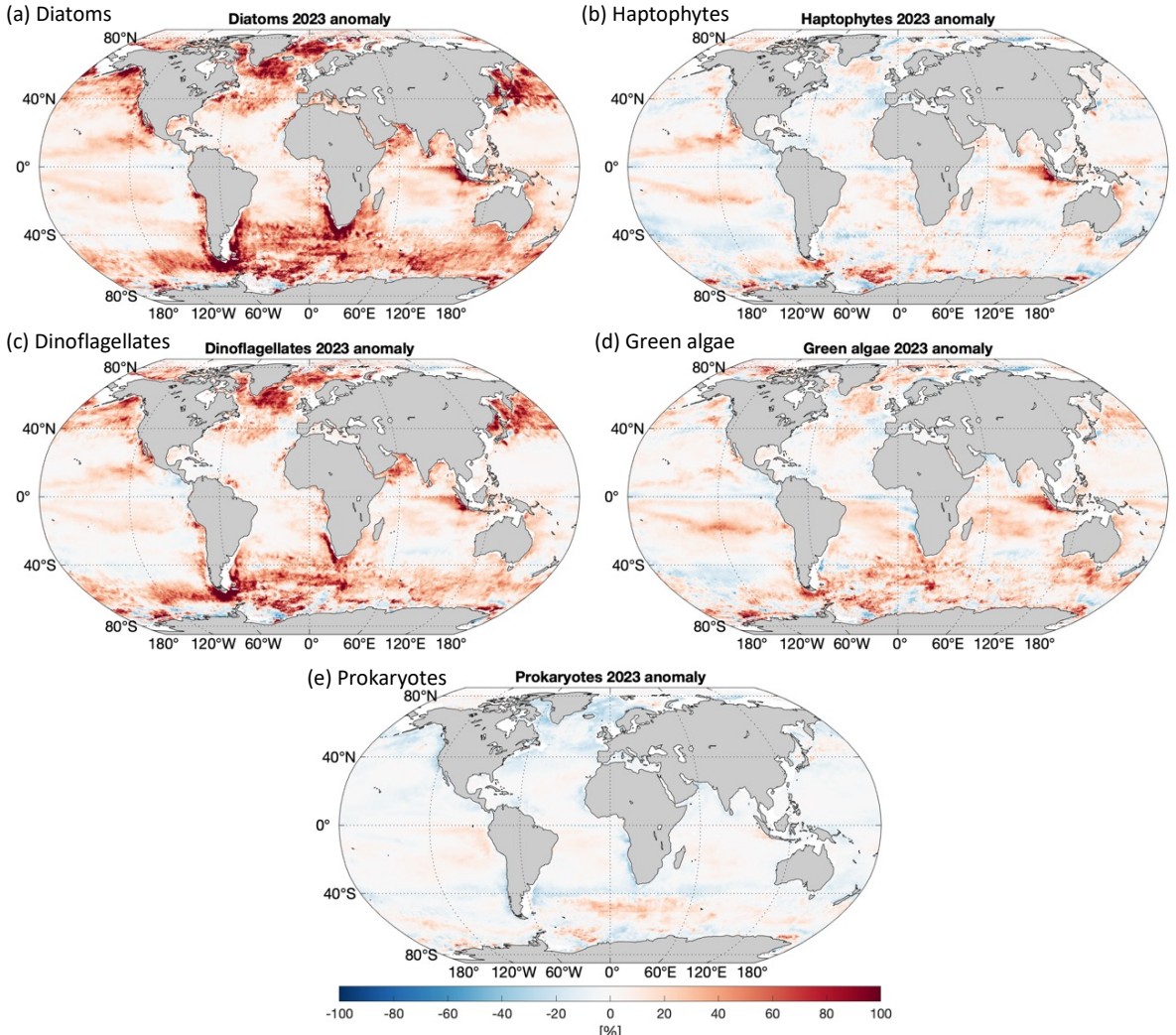

**Figure 6. Relative anomaly of 2023 for Chl*a* of (a) diatoms, (b) haptophytes, (c) dinoflagellates, (d) green algae, and (e) prokaryotes.**

## 4 Discussion, conclusions and outlook

### 4.1 The need for harmonization

Generating long term consistent PFT data from a single set of sensor(s) is challenging due to discontinuous satellite missions and different sensor specifications. PFT data derived using models established based on different sensor sets bear different levels of uncertainties. OLCI being the newest sensor has more spectral bands which should be beneficial for PFT retrievals, however, due to limited in situ pigment data set available for the model training, it does not show superior performance than the merged OC products. Harmonization is so far necessary for the current derived PFT products on the Copernicus Marine Service as it is not yet possible to produce consistent long-term PFT products using harmonized radiometric data from historic and current sensors using the proposed approach which requires more bands. Attempts have been carried out for consistent PFT products derived from big data driven deep learning ensembles by incorporating Rrs at only 5-6 merged bands, together with other ocean color and physical/biogeochemical variables (e.g., Zhang et al. 2024), which shows potential to upgrade the operational data sets, however the applicability of the implementation of such approach for operational products is yet to be testified.

### 4.2 MLBE correction scheme

This study aims to demonstrate consistent PFT time series data on the global scale and for the polar regions and the European Seas which were developed based on a robust machine learning correction scheme. The proposed MLBE correction scheme

outperforms the previously proposed method that was based on type II linear regression with considerations of PFT uncertainties (Xi et al., 2023a). For the overlapping period, the MLBE scheme demonstrates high consistency of the corrected OLCI-derived PFTs to the merged sensor derived PFTs, both in space and time, increasing our confidence in employing the data for further time series studies.

However, the MLBE model training was based on 12 months satellite data spanning only one year (the overlapping period of the two sensor sets), trying to identify the spatial variation of the PFT data from the two sensor sets, so that it could fit one pattern to the other on the whole global scale. It has been reported that random splitting between training and test sets may produce data leakages (Meyer et al., 2018; Stock et al., 2023) potentially leading to overoptimistic test performance that does not generalize well in actual application to other data sets. To avoid data leakage, temporal partitioning has been suggested to
ensure that the training and test data sets are independent. However, a random split was applied in the study as temporal partitioning was impractical due to the limited duration of the dataset in our case. The MLBE model is basically a correction scheme trained based on all pixel data (over 50 million available data points) from 12 monthly PFT products. The purpose was to cover as completely as possible the global region to ensure that the training learns the pattern globally. By applying the suggested temporal partitioning we would lose data, e.g., in high latitudes, if we exclude a certain month in the training. This
can cause biases in the learning process, then the trained model would very likely not be applicable to either the test set or other data sets that contain the missing periods. The straightforward random splitting in our study ensured the homogeneous splitting between the training and test data sets over space and time thanks to the large amount of data points, so that the trained model learned the most knowledge from the available data within the limited time period. Though such random partitioning has been widely used (e.g., Li. et al. 2023; Zoffoli, et al. 2025), one should keep in mind that having data for only a single year
is challenging because the year may present conditions that are specific to that year only which may cause unrealistic predictions for other years. It is therefore noteworthy that target-oriented data splitting and cross-validation such as considering spatial and temporal blocks should be applied in machine learning based studies when data set allows (e.g., Zhang et al. 2024).

For the next cycle of the implementation to Copernicus Marine Service, updates will be necessary for the PFT retrievals and the MLBE scheme. It is expected that the VIIRS-SNPP drifting after 2017 is better calibrated with the new reprocessing so
that our data used for the training in the correction scheme can be extended to more recent years to achieve an even better consistency between the merged sensor-derived and OLCI-derived PFT products.

4.3 Consistent PFT time series and validation

The time series generated based on the consistent PFT data on the global scale from 2002 to 2023 has shown a clear increasing trend for diatoms and dinoflagellates, and a slight decreasing trend for haptophytes and prokaryotes, while the green algae
exhibit no significant trend but with higher interannual variability. To date, the longest time series for ocean color products still covers less than three decades (starting in 1997 with the launch of SeaWiFS). Though it may still not be long enough for a robust trend analysis due to too strong decadal variability (Henson et al. 2010; Henson et al., 2016), these time series can help to catch distinct changes on different scales by comparing to the climatological state. Indeed, the findings such as significant increase in diatoms particularly after 2017 are of interest to in-depth investigations linking climate drivers to such
prominent changes. For instance, potential responses of phytoplankton biomass to increasingly frequent marine heat waves in the past years can be a suitable starting point.

Changes in phytoplankton biomass have been described by the Chl$a$ concentration derived from ocean color satellites covering the last decades. Trends of the Chl$a$ at different scales can be generated using current operational chlorophyll products such as OC-CCI and GlobColour. For instance, Chl$a$ as an Ocean Monitoring Indicator (OMI) has been included by the Copernicus
Marine Service where the climate trends of various OMIs are provided to indicate the state of the ocean health. The current published time series of Chl$a$ shows in general an increase during 1997–2022 on the global scale and also for the North Atlantic

and Arctic regions. The published per-pixel Chl*a* trend map shows a more prominent increasing trend in high latitudes but a slight decrease in mid- to low latitudes (e.g., https://doi.org/10.48670/moi-00230). These trends are in good agreement with our PFT time series which shows an overall increasing trend of the total biomass mainly due to the increased diatom biomass. Similar findings on both global and regional scales have been reported by Van Oostende et al. (2023), where the OC-CCI dataset has been used but with careful consideration of the spatiotemporal coverage of the different sensor datasets by applying a temporal gap detection method. Other techniques such as gap filling and statistical temporal decomposition are also in demand for more robust PFT data analysis, to increase the accuracy in separating the long-term signal from the seasonal component of the time series. Nevertheless, studies have shown that the OC satellite derived surface Chl*a* concentration presents contrasted trends between available products that are generated based on different retrieval algorithms and merging methods, e.g., the OC-CCI and GlobColour products (Yu et al., 2023), suggesting the need for careful interpretation of the trends for multi-OC sensor derived products. Inconsistencies between missions remain a significant challenge to overcome in order to provide climate-quality time series, which needs efforts from both the spatial agencies and scientific communities for correcting the inconsistencies in radiometric data with long-term time series and applying proper harmonization to the merged products (Pauthenet et al., 2024).

So far there are limited studies that investigated or reported the PFT interannual variability covering the recent years. There is also quite limited long term in situ PFT data available over large scales. However, our recent investigation at a smaller scale in the Fram Strait (Xi et al. 2024), has indicated that the surface diatom from the in situ data collected in the LTER Hausgarten area (75°N to 80°N, 5°W to 10°E) since 2009 has shown unanimous pattern with the satellite PFT time series, i.e., diatoms have shown an overall increase in this region in more recent years (satellite from 2018 but in situ from 2019 due to lack of data in 2018). The other PFTs show rather an oscillational feature but not as dramatic as seen in diatoms. It should also be noted that the in situ data were collected mostly in the spring to summer months (which vary from May to September) and cannot fully represent the phytoplankton development during the whole season or the interannual variabilities. However, these Fram Strait in situ data support our satellite time series with the diatom increase in the years from 2018 to 2023 in the Artic region. More field observations on phytoplankton community composition are constantly collected for further evaluations and hypothesis verifications.

Validation has been performed at different levels from the model development stage (details not shown in this study) to the corrected OLCI-derived PFT data to understand well the reliability of the datasets. Using validation data covering different times and regions, we observed that the satellite PFT data have larger discrepancies compared to the in situ data in high latitudes, especially in the Arctic Ocean, which has also been reflected in the per-pixel uncertainty assessment for the EOF-PFT algorithm (Xi et al., 2021). Compared to the original OLCI-derived PFT data, the MLBE corrected data showed comparable but unimproved validation statistics against the in situ data sets, which can be explained by the following aspects: 1) limited temporal coverage of the training data used in the MLBE might transfer further errors to the corrected data. 2) Data set 1 served as the test set randomly extracted from the global in situ data set from which the other 70% has been used to train EOF-PFT model for the OLCI sensors, therefore the data set 1 possessed similar features with the training set and exhibited the best agreements with the OLCI-derived PFTs before correction, very possibly due to the aforementioned data leakage effect. 3) The MLBE scheme bears lower correction efficiency in high latitudes due to larger inherent uncertainties in the satellite-derived PFT products. However, our validation for diatoms and dinoflagellates in the Arctic Ocean using dataset 2, collected during 2021–2023, shows no overestimation of the satellite retrievals compared to the in situ data despite the weak correlation and higher discrepancies (Figure 3b), indicating that our satellite retrievals presented correctly the increased biomass for the two PFTs. Since the ecosystem in the Arctic Ocean undergoes fast changes as a consequence of the arctic warming and sea ice retreat, there are still a lot of unknowns on how the phytoplankton community adapts and responds to these changes (Oziel et al., 2018; Meredith et al, 2019). It is potentially essential for the Copernicus Marine Service to provide not only for the white

ocean (sea ice) but also for the green ocean (biogeochemical parameters) a wide range of biological/biogeochemical variables to better understand the state and possible tendencies of the ecosystems in the Arctic Ocean.

## 4.5 Conclusion and outlook

The correction scheme proposed in this study is specifically designed to address inter-sensor data inconsistencies in the current Copernicus Marine Service PFT products. The present trained model can only be used to correct the OLCI-derived PFT product to match the merged sensor-derived product. However, the underlying technical framework is adaptable to other common ocean color products, such as optical properties derived from multiple sensors, thereby enhancing the overall continuity and consistency of ocean color data. As a rapidly emerging and powerful technique, machine learning can be further leveraged in ocean color data services, supporting agencies and data platforms in delivering high-quality, consistent operational products. This work is at the cutting-edge attempting to demonstrate the most up-to-date long-term phytoplankton community in several functional groups derived from ocean color products. Providing inter-annual variation and trend analyses of the surface phytoplankton community structure, the PFT products will complement the chlorophyll products on the Copernicus Marine Service as an essential ocean variable and help in the assessment of the ocean health in the biogeochemical aspect.

**Data availability**

Data and products used in this study and their availabilities and supporting documentations are listed in Table 1, from which the in situ HPLC pigment concentrations and the corresponding derived in situ PFT Chl*a* data used for validation are published on PANGAEA (https://doi.org/10.1594/PANGAEA.982433, in review).

**Author Contributions**

HX, AB, MB and AM conceptualized the study. HX designed and carried out the experiments. MB and JD provided support in satellite products and matchup data extraction. EM contributed to the machine learning algorithms. HX drafted and revised the manuscript with contributions from all co-authors.

**Competing interests**

The authors declare that they have no conflict of interest.

**Acknowledgments**

We thank the two Copernicus Marine – Innovation Service Evolution R&D Projects, GLOPHYTS (2022-2024) and ML-PhyTAO (2024-2026), for funding. Copernicus Marine Service is implemented by Mercator Ocean International in the framework of a delegation agreement with the European Union. This work was also partly supported by DFG (German Research Foundation) Transregional Collaborative Research Center ArctiC Amplication: Climate Relevant Atmospheric and SurfaCe Processes, and Feedback Mechanisms (AC)3 (Project C03), and the ESA project 4DMED-Sea (4000141547/23/I-DT). Ehsan Mehdipour's work was supported by the project "4D-Phyto" funded by AWI-INSPIRES and HGF-MarDATA. Thanks to ESA, EUMETSAT and NASA for distributing ocean color satellite data, and especially to the ACRI-ST GlobColour team for providing OLCI and merged ocean colour L3 products. In situ data from four *Polarstern* expeditions were funded under Grants AWI_PS126_02, AWI_PS131_5, AWI_PS133/1_11 and AWI_PS136_04, respectively. Captain, crew, and expedition scientists are also acknowledged for their support at sea. We also address our acknowledgements to Dr. Alexandre Castagna and the other reviewer for their constructive comments in improving this study.

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
