# Peer review of "Consistent long-term observations of surface phytoplankton functional types from space"

_State of the Planet, 2024_

## Author Comment (AC1)

**Consistent long-term observations of surface phytoplankton functional types from space**

Hongyan Xi[1*,] Marine Bretagnon[2], Ehsan Mehdipour[1,3], Julien Demaria[2], Antoine Mangin[2], Astrid Bracher[1,4]

[1]Alfred Wegener Institute, Helmholtz-Centre for Polar and Marine Research, Bremerhaven, 27570, Germany
[2]ACRI-ST, Sophia Antipolis Cedex, France
[3]School of Business, Social & Decision Sciences, Constructor University, Bremen, Germany
[4]Institute of Environmental Physics, University of Bremen, Bremen, 28359, Germany

*Correspondence to*: Hongyan Xi (hongyan.xi@awi.de)

**Author Comments in response to Referee #1**

**Overview**

The authors discuss that changes in phytoplankton biomass, proxied by chlorophyll *a* concentration (Chla), partitioned between different phytoplankton functional types (PFTs) provides not only direct ecological information but also indirect information about environmental change due to niche differences between the functional types. PFTs in optical applications are groupings of phytoplankton separable by optical signatures of pigment composition (and possibly ancillary environmental information), that however contain a reasonable degree of taxonomical and ecological information. Such partitioning of bulk Chla can in principle be applied to remote sensing, and has been demonstrated globally for open ocean areas. However, due to the limited lifespan of orbital remote sensing missions, long term analysis based on remote sensing requires merging of data from multiple sensors, including harmonization to account for differences in sensor specification, calibration, and performance. Nonetheless, the official PFT product released by the Copernicus Marine Services (CMEMS) includes appropriate merging only for data between 2002 and Apr-2016, using data based only from the Sentinel-3 mission from May-2016. This resulted in a continuity issue in the time series of PFT products from CMEMS. The authors attempted a previous harmonization method they had developed for a study in a specific ocean region, but despite it achieving a good performance on the global average, spatially resolved data over the globe showed large discrepancies between the datasets on regional scales. Therefore, the authors developed a new harmonization method that also takes as input the spatial information, in the form of geographic coordinates, in order to harmonize the data on a spatially resolved level, and the global average as a consequence. This allowed the authors to evaluate a 2 decade long time series of PFT abundance at the global average and over selected regions. The authors then describe the temporal patterns observed and provide some comment concerning the use of this information.

The study represents further progress in an relevant research topic, in which the consortium of authors includes recognized experts and leader in the field. It provides a method that potentially improves the official PFT CMS products, and provides new information on temporal and spatial changes of optical/ecological phytoplankton groups. The written and visual presentation is mostly clear, with exceptions noted in the detailed review below.

While in general my perspective is positive, I do have concerns related to the methodology and results that can impact the interpretation in the study and I believe should be addressed for a publication. My comments are divided in three sections: Major comments and Minor comments, have comments that would be expected to be addressed by the authors for a publication, and Suggestions, which contain suggestion for improvement that need not be addressed by the authors.

We thank the reviewer very much for providing us constructive and elaborate comments and suggestions, which help a lot to improve the quality of the manuscript. We have carefully addressed these the points in our response below to each comment and will revise the manuscript accordingly as indicated in the responses.

**Major comments**

1. **Source data for PFT products and the need for harmonization at the PFT product level**. Section 2.1 provides the description of the database used for the PFT products, indicating that GlobColour merged product is used from Jul-2002 to Apr-2016 and OLCI data used from May-2016 to present. The justification to not use the GlobColour merged product for the whole dataset period (Jul-2002 to present) is that VIIRS/SNPP data has been identified to contain biases possibly caused by sensor degradation. I have little experience with GlobColour or the CMEMS PFT product, so it is possible I'm missing something, but I do not find that justification to be sufficient, specially considering that among the authors are the producers of the GlobColour merged dataset. For example, is it the case that GlobColour cannot exclude VIIRS/SNPP data since 2017 from the merged product calculation due to some specific reason? Or is a solution being sought but not implemented in the time frame of this study? This is relevant because if the radiometric data, base for the PFT calculation, is not harmonized between sensors, then the PFT product will have to be harmonized, which is the topic of this study. And since Xi et al. (2023) have already dealt with such harmonization, the problem seems to be known for a few years. To be clear, I believe it is fair that the study took this course, but I'm missing more discussion on the why it was so. For example, regardless of the situation with GlobColour, as discussed above, this course could be justified if PFT derived from OLCI/Sentinel-3 alone would be superior (due to higher spatial resolution and more bands) but that does not seem to be the case as PFT data derived from OLCI/Sentinel-3 data is corrected to match the PFT data derived from the merged dataset. The authors could argue that the way forward is merging of the derived product (e.g., PFT) and not of the base radiometric data as the configurations of the new sensors become more diverse, and so the future of initiatives such as GlobColour is the merging of derived products. Again, I may be missing something, but it seems to me a clearer justification is necessary.

   We respond to this comment from the following aspects:

   • GlobColour merged products on the Copernicus Marine Service

   GlobColour (https://hermes.acri.fr/) provides merged ocean color (OC) products and also products from single sensors, however, only all sensor merged OC products and Sentinel 3A/B OLCI products from GlobColour have been implemented to the Copernicus Marine Ocean Data Store through the Copernicus OC-TAC (Ocean Color Thematic Data Assembly Center) consortium, consistent with another version ESA-CCI (OCCCI) products. Therefore, two datasets are supplied by GlobColour on the Copernicus Marine Service:
   1. Merged products (all available OC sensors including OLCI) with the longer timeseries;
   2. OLCI products with a better spectral resolution and higher frequency of revisit (sum Sentinel 3A OLCI and + Sentinel 3B OLCI).

   Our PFT products for the Copernicus Marine Service are meant to be derived from the merged OC products and OLCI data too, to keep the consistency of the source data. However, we had to include more bands (at least eight) for Rrs data than that are available on the Copernicus Marine Service (which is five merged bands Rrs) to train our PFT algorithms, so we used the merged Rrs data from GlobColour instead which provides more available bands from SeaWiFS, MODIS, MERIS, and VIIRS SNPP sensors (either merged or single sensor).

- VIIRS SNPP data

We understand the reviewer's point that one should avoid using products that are sought to be biased due to sensor drifting. GlobColour takes directly the VIIRS data with the NASA (re)processing, which is the same as for the MODIS Aqua data. Such drifting in the VIIRS data after 2017 could not be resolved before we performed our study. However, we could not exclude VIIRS data because our PFT retrieval algorithm was developed based on the eight bands Rrs from the merged sensors which after 2012 are from MODIS and VIIRS merged data. We had to include VIIRS to generate a complete PFT product time series from 2002 onwards, and due to the drifting of VIIRS we can only choose one year overlapping with OLCI in this study.

Good news is that lately the GlobColour has implemented the latest reprocessing (R2022.0) where the VIIRS SNPP drift has been corrected, and its time series shows good consistency with other sensors such as MODIS and VIIRSJ1 as highlighted by the NASA monitoring (https://oceancolor.gsfc.nasa.gov/data/analysis/global/). With that we can investigate (but not in this study) a longer overlapping period by reproducing all the relevant PFT products and check the validity of the approach.

- Harmonization

The inconsistency between PFT estimates from merged OC sensors and OLCI are inherent because two sets of models have been used, even though both are empirical orthogonal function based (Xi et al. 2021), they are based on different spectral bands information, trained with two different satellite-in situ matchup data sets spanning different time periods with different amount of matchup points, i.e. for merged OC sensors we obtained ~1500 matchup data points from 2002 to 2016, and for OLCI only ~ 300 from 2016 to 2021 which is simply due to the shorter time span and fewer data being publicly available for the most recent years. Though the algorithm is based on more spectral information because more bands are available from OLCI, our uncertainty analysis following Xi et al (2021) has revealed that the OLCI derived PFTs bear higher per-pixel uncertainty than the PFTs from the merged OC sensors. This is possibly due to the smaller training data set. Therefore, we decided to correct OLCI derived PFTs to the merged OC sensor derived PFT for deriving a consistent data set. Our first attempt was the straightforward linear regression correction as proposed in the Xi et al. (2023) study. However, this method is ineffective in reducing the spatial discrepancies. Therefore, we finally employed in this study the random forest regression, among several ML methods tested, to resolve the spatial inconsistency, which turns to perform much more effective in harmonization of the two types of PFT data sets. Such harmonization is necessary at least for the current derived PFT products on the Copernicus Marine Service, as it is yet not possible to produce consistent long-term PFT products using harmonized radiometric data from historic and current sensors. Attempts have been carried out for consistent PFT products derived from machine learning ensembles trained by incorporating Rrs at only 5-6 merged bands, other ocean color and physical/biogeochemical variables (e.g., Zhang et al. 2024), which probably would help to upgrade the operational data sets, and such a harmonization might not be necessary anymore. The applicability of such an implementation is yet to be testified. However, in our previous work (Xi et al. 2020) it was shown that a minimum of eight bands for the EOF-based PFT algorithms is necessary to produce data sets of high accuracy.

We agree with the reviewer that more explanation is necessary – and due to the length limit – we have added a short discussion as following:

"Generating long term consistent PFT data from a single set of sensor(s) is challenging due to discontinuous satellite missions and different sensor specifications. PFT data derived using models established based on different sensor sets bear different levels of uncertainties (Xi et

al., 2020). OLCI, being the newest sensor, has more spectral bands which should be beneficial for PFT retrievals. But, due to limited in situ pigment data set available for the model training, it does not show superior performance than the merged OC products. Harmonization is so far necessary for the current derived PFT products on the Copernicus Marine Service as it is yet not possible to produce consistent long-term PFT products using harmonized radiometric data from historic and current sensors. This is because the EOF-based algorithms require at least eight bands (Xi et al. 2020). Attempts have been carried out for consistent PFT products derived from big data driven deep learning ensembles by incorporating Rrs at only 5-6 merged bands, together with other ocean color and physical/biogeochemical variables (e.g., Zhang et al. 2024), which shows potential to upgrade the operational data sets. Though, the applicability of the implementation of such approach for operational products is yet to be testified."

2. **Overlapping period for model calibration and partitioning of data for model calibration and validation**. Section 2.1 describes that though the official PFT CMEMS product uses the merged GlobColour product only from 2002 to Apr-2016, this was extended by the authors to Apr-2017 in order to provide a year of overlap with PFT from OLCI/Sentinel-3 (May-2016 onwards) for model calibration and validation. It is possible that at least a year of overlap is necessary to fully take in account the variability caused by seasonality, but having a single year is challenging as that specific year may present conditions that are specific to that year only. Considering the argument for not using the merged product beyond 2017 discussed in the previous point, I understand that one year is all data that it is available, however, the potential impacts of this single overlap year should be discussed. A closely related issue is the data splitting between calibration and validation sets described in Section 2.2. A random subset can produce "data leakages" (information in the validation set also present in the calibration set) that produce overly optimistic performance statistics (e.g., Meyer et al., 2018). In the case of this study, the model application (May-2016 to present) is beyond the time domain of the model calibration (May-2016 to Apr-2017), so in my view an ideal validation strategy should demonstrate that the model can well extrapolate in time, which is not possible when the data is randomly split as the calibration and validation sets cover the same time period. I note that this is specially necessary because while before (regression method of Xi et al., 2023) the correction was a matter of relating two variables over a large spatial domain, now the relation depends explicitly on geographical location, for which dynamics might be changing in time. This is somewhat a conundrum considering again the data availability for the GlobColour merged product as it is (with potentially bias data from VIIRS/SNPP from 2017) and the potential need to include all seasons. Nonetheless, I believe those aspects to be very relevant and that they should be considered in the discussion.

We appreciate the reviewer's suggestion and the concern about 'data leakage'. We understand the logic and reason why the inclusion of temporal partitioning is important. However, it is uneasy to apply it to our case here. Our MLBE model is basically a correction scheme, that is trained based on 12 months satellite data spanning only one year (the overlapping period of the two sensor sets), with the model we wanted to set up a regression model through random forest learning trying to identify better the spatial variation of the PFT data from the two sensor sets, so that it could fit one pattern to the other on the whole global scale. We considered all pixel data (over 4 million available data points) from the 12 monthly products, and we wanted to cover as complete as possible the whole global region to make sure the training learns the pattern globally. By applying the suggested temporal partitioning we would lose data, e.g., in high latitudes, if we exclude a certain month in the training. This may cause biases in the learning process. Then the trained model would very likely not be applicable to the test set (because though they would be temporally

independent from the training set, the spatial information which is not included in the training set could not be learnt from the training and thus the ML correction model might fail in the test data and also in the global products in later years). Though we applied a straightforward random splitting in this study, the training and test data sets were nearly homogeneously divided over space and time by the random splitting due to large amount of the data points (> 4 millions), as shown in Fig. R1 the cumulative distribution function (CDF plots for each of the input variables). This makes sure that the trained model take the most knowledge of the available data sets within the limited time period that can be used in the correction model.

[Figure]

*Figure R1. Cumulative distribution functions of input variables (PFT, lat, lon) involved in the MBLE training set, taking diatom as representative.*

We agree to the reviewer's comment, as the test set carries very similar information as the training set, which can produce overly optimistic performance statistic, and having data for only a single year is challenging because the year may present conditions that are specific to that year only which can cause wrong predictions for other years.

Reviewer 2 also posted a similar comment and we understand that this is a critical point. However, this potential 'data leakage' could not be well resolved in this study due to limited overlapping data set. We hence add a paragraph below about 'model caveats' in the discussion to cover this aspect:
"However, the MLBE model training was based on 12 months satellite data spanning only one year (the overlapping period of the two sensor sets), trying to identify the spatial variation of the PFT data from the two sensor sets, so that it could fit one pattern to the other on the whole global scale. It has been reported that random splitting between training and test sets may produce data leakages (Meyer et al., 2018; Stock et al., 2023) which result in overoptimistic performance in the test data but less good performance in actual applications to other data sets. To avoid data leakage data temporal partitioning has been suggested to ensure that the training and data sets are independent. However, random split was applied in the study as the temporal partitioning does not apply to our case. The MLBE model is basically a correction scheme trained based on all pixel data (over 50 million available data points) from 12 monthly PFT products. The purpose was to cover as complete as possible the global region to ensure that the training learns the pattern globally. By applying the suggested temporal partitioning we would lose data, e.g., in high latitudes, if we exclude a certain month in the training. This can cause biases in the learning process, then the trained model would very likely not be applicable to either the test set or other data sets that contain the missing periods. The straightforward random splitting in our study ensured the homogeneous splitting between the training and test data sets over space and time thanks to the large amount of data points, so that the trained model learned the most knowledge from the available data within the limited time period. Though such random partitioning has been widely used (e.g., Li. et al. 2023; Zoffoli, et al. 2025), one should keep in mind that having data for only a single year is challenging because the year may present conditions that are specific to that year only which may cause unrealistic predictions for other years. It is therefore noteworthy that target-oriented data splitting and crossvalidation such as considering spatial and temporal blocks should be applied in machine learning based studies when data set allows (e.g., Zhang et al. 2024)."

3. **Reference PFT from in situ data**. Though Section 2.3 provides information about the different datasets, their sampling and partitioning, no information is given on how PFT Chla is calculated from in situ measurements. The authors have addressed this important issue in previous publications, but a summary of the method should also be included here, with citations to their previous work where the methodology is discussed in greater detail. In addition, any changes to methods or to the base data analysis methods (instruments, protocols, operators, etc) for calculation of PFT from in situ measurements should be noted, and if it is the case, discussed if they may impact the validation in Figure 3b (dataset 2, Section 3.2).

Thanks for pointing out this missing information. The PFT calculation for the validation data sets is consistent with that used in the algorithm retuning for the updated version, which has been implemented to generate PFT products for Copernicus Marine Service in Nov. 2024. We have added in the revised manuscript in Section 2.3 "The in situ data were derived from quality-controlled in situ HPLC pigment concentrations using the diagnostic pigment analysis (DPA) with updated pigment-specific weighting coefficients following Xi et al. (2023a; 2023b), consistent with the calculation of the in situ PFT data used for the updated EOF-PFT algorithms described in Section 2.1."

4. **Calculation of spatial averages**. The methods do not specify how the regional and global averages were calculated, in particular if the area contribution of each pixel was compensated for the area distortion of a regular lat/lon grid. It is possible that this was taken into account, but since it is not mentioned in the manuscript I ask the authors to confirm. This is mostly relevant for the global average, but also applicable to the regional averages. In my understanding, the average should contain a weight proportional to the cosine of the latitude to make the contributions of each pixel to the average proportional to their contributions to the surface ocean area. This would likely change considerably the global trend in diatom Chla, as the increase seem to be driven by increases at high latitudes.

We agree with the reviewer about the area distortion when calculating the mean spatially, and a latitudinal weighted average can compensate the geographical distortion by taking into account proportional contribution. We therefore have modified the calculations of the global and the four regional averages when generating PFT time series. The latitude-weighted averaging was applied to the logarithmic transformed Chl*a* concentrations to get the log based mean which are then converted to their natural values. So for each monthly product over a certain region, the average was calculated based on the equation below:

$$Mean_{chla\_diatom} = \exp\left(\frac{\sum \cos(lat) \cdot \ln(Chla_{diatom})}{\sum \cos(lat)}\right)$$

We have added the following text in the revised manuscript: "PFT time series of different spatial scales were calculated by applying the weighted average (taking cosine of the latitude as weights) to the monthly PFT data over the defined regions, to take into account the proportional contribution of each pixel to the global surface ocean due to area distortion in the gridded dataset. The latitude-weighted averaging was applied to the logarithmic transformed PFT Chl*a* to get the log-based mean which are then converted to natural values."

The time series plots in Figure 4 have also been updated, showing overall slight changes in the trends, however the Chl*a* magnitudes of the PFT time series at global scale are in general slightly lower for the weighted average, except for prokaryotes. This is mainly due to much lower prokaryotes Chl*a* concentrations in high latitudes (compared to lower latitudes) contributed less with the latitude weighting applied, contrasting to other PFTs which have in general higher Chl*a* in higher latitudes. The trend of the global diatom Chl*a* was slightly decreased (from 0.0014 to 0.0011 mg m-3 per decade), while their increase at high latitudes is still very prominent, as the proportional weights are also considered in the 'divider' which is the weighted total number of observations. Accordingly, we have updated Figure 4 and the statistical description related to this figure in Section 3 of the revised manuscript.

[Figure]

**Figure 4. Panel (a): Updated (corrected) time series of the five PFT Chl*a* based on the global mean from 2002 to 2023. Merged products cover the period of July 2002-April 2017 (indicated with dots), and OLCI products are for May 2016-Dec 2023 (indicated with crosses). Note that the OLCI products have been corrected to merged products based on MLBE. Panel (b): Trends of diatoms, haptophytes, dinoflagellates, green algae and prokaryotes Chl*a* on the global scale and four regional scales (the North Atlantic Ocean, the Mediterranean Sea, the Arctic Ocean and the Southern Ocean), respectively. Trend slopes per decade with uncertainties have been indicated with significant trends marked with an asterisk (*).**

5. **Temporal pattern shift after the overlapping period**. Despite the efforts in data harmonization, the limitations of the cal/val procedure discussed before and the validation analysis for high latitude data ask for additional support for the temporal patterns observed after the overlapping period. For example, the Chla of diatoms changes considerably from 2017 onwards (Figure 4). A corresponding change in behaviour is observed for green algae and prokaryotes. Maybe the authors can provide independent evidence sourced from the literature (specially in situ studies, if any is available) for the temporal / regional patterns they observe in given PFT, in order to provide support that the changes are not potentially influenced by the correction procedure. The two external supports provided by the authors (CMEMS trend product and the study of Van Ostend et al., 2023) concern only bulk Chla, that is, without information about increasing or decreasing abundances of specific groups. This is also relevant specially considering that validation in Figure 3b suggests poor performance of the retrievals against in situ data for the Arctic. This performance results should be considered in the discussion of the global average and Arctic average trends - currently the discussion states only, and in my opinion incorrectly, that no overestimation was observed for the validation with Arctic data in dataset 2.

We have noticed the insufficient discussion regarding the PFT trends or changes. So far there are limited studies which investigated or reported the PFT interannual variability covering the recent years. There are also quite limited long term in situ data over large scales. However, our recent investigation at a smaller scale, i.e., in the Fram Strait, has indicated that the surface diatom from our in situ data collected in the LTER Hausgarten area (75°N to 80°N, 5°W to 10°E) since 2009 has shown unanimous pattern with the satellite PFT time series (Figure R3 and R4) – i.e., diatoms have shown an overall increase in this region in more recent years (satellite from 2018 but in situ from 2019 due to lack of data in 2018). The other PFTs show rather more an oscillational feature but not as dramatic as seen in diatoms. It should also be noted that the in situ data were collected mostly in the spring to summer months (which vary from May to September) and can not represent fully the whole season. However, the Fram Strait in situ data support our satellite time series showing the elevation of diatoms in the last five years until 2023 in the Artic region. These results have been presented on the Ocean Optics Conference as a poster in Oct 2024 (Xi et al. 2024).

[Figure]

[Figure]

*Figure R2. Time series of satellite derived total Chla (upper panel) and PFT Chla (bottom panel) in the Fram Strait.*

*Figure R3. Boxplots showing in situ TChla and PFT Chla time series from 2009 to 2023. PFT Chla are derived from HPLC pigment data using diagnostic pigment analysis (DPA) following Xi et al. (2023a,b). Data were collected from the LTER 'HAUSGARTEN' expeditions in the Fram Strait: PS74, PS76, PS78, PS80, PS85, PS93.2, PS99, PS106/107, PS121, MSM93, PS126, PS131, PS136, available from Xi et al. (2023b) except for the last four expeditions.*

We have added a brief discussion regarding this point - "So far there are limited studies which investigated or reported the PFT interannual variability covering the recent years. There are also quite limited long term in situ data available over large scales. However, our recent investigation

at a smaller scale (Xi et al. 2024) in the Fram Strait, has indicated that the surface diatom from the in situ data collected in the LTER Hausgarten area (75°N to 80°N, 5°W to 10°E) since 2009 has shown unanimous pattern with the satellite PFT time series, i.e., diatoms have shown an overall increase in this region in more recent years (satellite from 2018 but in situ from 2019 due to lack of data in 2018). The other PFTs show rather more an oscillational feature but not as dramatic as seen in diatoms. It should also be noted that the in situ data were collected mostly in the spring to summer months (which vary from May to September) and cannot represent fully the whole season. However, the Fram Strait in situ data support our satellite time series showing the elevation of diatoms in the last five years until 2023 in the Artic region."

6. **PFT anomaly in 2023**. I'm not convinced this analysis add information to the study. I would argue that in principle a given year anomaly is relevant in a specific study about that year phenomena, but in the context of global change, a given year anomaly would seem relevant to me only if the anomaly is beyond a given range of variability of the data along the 2 decade of the climatology (e.g., 2 standard deviations).

We understand the reviewer's point. We make some clarifications here regarding the content of the manuscript. This study was conducted as a contribution to Chapter 2 "Updated and new pathways in ocean science" of the 9th Edition of the Copernicus Ocean State Report, where Essential Ocean Variables and Ocean Monitoring Indicators should be included to build the baseline for the scientific studies. Results for the global ocean and for European regional seas should be included, and should cover information over the past decades, and up to the target year of each Report cycle which is 2023 for the OSR9 (submission in 2024, final release in 2025). With these guidelines, it is necessary to include the state of the PFTs of the latest year (i.e. 2023) into our study as an update, and anomaly maps presents more meaningful information rather than the plain PFT distribution maps. Though the focus of this study is not about this specific year phenomena, it is still worthwhile to show that certain PFTs e.g. diatoms of 2023 are dramatically enhanced in higher latitudes compared to the average state of the last two decades. We hence still keep the anomaly maps in the revised manuscript.

7. **Relative composition**. While the study provides the analysis in terms of Chla as a proxy for biomass of each PFT, one could expect that new information is available when normalizing the data to evaluate changes in relative composition. For example, even though diatoms might be increasing in the a given environment, other groups might be increasing faster and so the relative contribution of diatoms be decreasing. I believe such analysis would add more to the current study than the 2023 anomaly analysis commented above. However, such analysis is not necessary and the study is sufficient without it.

We have also thought about changes in the relative composition of PFT to the total biomass and have looked at the fraction normalized to the total sum of the PFT Chla, assuming that the sum of the 5 PFTs representing the majority of marine phytoplankton (Figure R2). There are still other groups such as cryptophytes and pelagophytes, considering there relatively low contribution to the total biomass (Zhang et al. 2024) we assume they don't affect significantly the general pattern of the PFT composition time series. Ideally, we could also use total Chla product as the base knowledge. However, due to discrepancy between the Chla products derived from different processings and algorithms such as OCCCI and GlobColour (Brando et al. 2024; Garnesson et al. 2024), the PFT fraction estimates from different product can also be quite different (this aspect is however out of the scope of this study). Because of this, we chose to simply use the sum of the five PFTs for the fraction estimation and as a showcase the global PFT fraction time series are been shown here. We

decided to include the global PFT fraction time series in this response only, as it will also be publicly available after the review process. We would not like to show such plots in the manuscript due to length limit of the manuscript and also because they present more or less similar information with that of the PFT Chl*a* time series: i.e., with an increase in diatom Chl*a* and a decrease in prokaryotes from 2017-2018, though some different temporal patterns have also been observed, such as haptophytes and green algae present a relatively stable contribution to the total biomass though their absolute quantifications showing more inter-annual variability (Figure 4 global time series).

[Figure]

*Figure R2. Time series of PFT fraction normalised to total sum of PFT Chla (legend .*

**Minor comments**

1. Abstract: The OLCI acronym is not defined when used in the abstract.

   This has been revised.

2. Line 16: "the merged sensor-derived PFT" - it not clear to me what the authors are referring to. The main text makes clear this is specifically the GlobColour merged dataset, one of the inputs to the CMEMS PFT product. While understanding the limitations of space in the abstract, the authors are encouraged to modify the abstract to make it more specific.

   We have revised the phrasing in the abstract as below and also clarified it in the introduction section due to space limitation of the abstract.
   "The correction scheme is applied to the Sentinel 3A/B Ocean and Land Color Instrument (OLCI) derived PFT data, to match them with the PFT data derived from GlobColour merged ocean color products using the overlapped period."

3. Line 45 and other locations: The study is described as merging PFT products from different sensors, but I understood that what is being merged is the PFT products from two datasets: PFT from GlobColour merged sensors and PFT from OLCI/Sentinel-3. In that sense, phrasing it as merging between sensors can cause confusion.

   We checked and rephrased all related terms throughout the text to avoid the confusion.
   ~ L38-41, 46-48 in the revised manuscript

4. Line 68: It seems the potential bias is identified specifically in the VIIRS/SNPP, and not on VIIRS/NOAA-20. If that is correct, please add the mission specification to the text. Exactly,

the GlobColour merged OC data include only SeaWiFS, MODIS, MERIS, and VIIRS-SNPP during their life time, VIIRS-J1/NOAA-20 is not included. It has been clarified in the revised version. ~ L38-41

5. Line 86: Please provide a citation (Xi et al., 2021? or "this study", if it is the case) for the statement that PFT from OLCI/Sentinel-3 carries higher uncertainty than PFT data from merged GlobColour.
Xi et al. (2021; 2023) were cited for the uncertainty statement.
A similar validation analysis to that presented in Figure 3 Section 3.2 could be provided for the OLCI data before harmonization to the GlobColour merged sensor data as supplementary material.
During the revision we realized that the validation procedure was not rigorous enough which might cause the poorer validation performance for the corrected data than that of the original OLCI-derived PFT. This I s due to that the MLBE scheme in this study was trained based on coarse resolution 25 km monthly products. This was the resolution used to produce the time series analysis including trends (see section 2.2). However, for finally using the daily 4 km PFT products, the MLBE scheme must be trained also on this temporal and spatial resolution which is necessary for producing the correct satellite matchup PFT data. Applying the monthly 25 km-based correction seems to result in inconsistent spatial projection e.g. from larger pixels to much smaller pixels. We are currently working on a higher resolution-based ensemble by applying the same MLBE concept as described in the manuscript and then re-generate the validation. However, the computation is quite heavy so it takes a bit more time. We expect that the updated validation should be comparable to the validation using the OLCI derived data before the correction (which are shown below).

We will ask the journal editor if it is possible to provide supplementary materials due to the limited space in such a scientific report. If this will not be possible, these validation results are also available now in this response document.

[Figure]

*Figure R3. Original PFT Chla derived from OLCI sensors in comparison with in situ PFT dataset 1 (global data set from 2016 to 2020, N=99).*

[Figure]

*Figure R4. Original PFT Chla derived from OLCI sensors in comparison with in situ PFT dataset 2 (only own expeditions in high latitudes from 2021 to 2023).*

6. Line 101: It is not clear to me what the PFT data transformation achieves. It is stated that the log-transform and scaling/shifting achieve a normal distribution in the rage [0, 1], but a normal distribution is not restricted to a finite range and shifting/scaling does not change the data distribution type, just the magnitude of its specific parameters (e.g., mean and standard deviation). Authors are encouraged to rephrase or provide further details. In addition, they are encouraged to provide more information on the transformation used to achieve latitude and longitude data in normal distribution and in the range [-1, 1].

We apologize for the confusing statement in the original manuscript. The geographic information (latitude and longitude) were just simply normalised to the the range [-1,1] given their original ranges of [-89.875, 89.875] and [-179.875, 179.875] (with 0.25° resolution) and they follow uniform distribution. They were not transformed to a normal distributed set. The PFT Chl*a* datasets, similar to the total Chl*a* product, basically follow the log-normal distribution. Therefore, they were just log-transformed in the emsemble training. As the purpose of the learning scheme is to correct OLCI-derived PFT data set to the merged sensor derived PFT data set, the scaling of the input log-based PFT to [0,1] is not really necessary. During this study we tried both scaling and non-scaling, the results showed no difference and ensemble used in this manuscript was actually generated with the non-scaling but log transformed PFT data.

Therefore we have modified this statement in the manuscript by revising the sentence "Before performing the training, latitude and longitude were normalised to [-1, 1], and natural log-transformed PFT data were normalised to [0, 1] so that these parameters are in Gaussian distribution" to "Before performing the training, the PFT data sets were log-transformed due to their nature of log-normal distribution (Xi et al. 2021). The geographic information (latitude and longitude) were simply normalised to the range [-1,1] by scaling their original ranges of [-89.875, 89.875] and [-179.875, 179.875] (with 0.25° spatial resolution)."

7. Line 123: The acronym CMEMS is used for the first time here, but not defined. Note that Copernicus Marine Service is mentioned for the first time in Line 42, where the acronym could be defined.

   We now use Copernicus Marine Service everywhere in the manuscript, as it is not recommended anymore by the Mercator Ocean International (MOi)* to use any acronyms such as CMEMS or CMS in reports, publications and presentations. It was emphasized specifically at the last Copernicus Marine Phase II Kickoff Meeting in Jan 2025 in Toulouse, so we will follow the new recommendation.
   *MOi is an intergovernmental organisation selected by the European Commission (EC) in 2014 to implement the Copernicus Marine Service.

8. Line 175: The authors evaluate that only diatoms, dinoflagellates and prokaryotes show some relation between sensor-derived and in situ PFT data in the validation analysis presented in Figure 3b. However, in my perspective the Arctic dataset shows no relation for any group except perhaps prokaryotes. This suggests that the Arctic dataset (or region) is the most problematic (not necessarily polar regions or high latitudes in general), and where future work could focus.

   We strongly agree with the reviewer. In the manuscript we have also emphasized that the future work should focus more on the high latitudes to provide a better PFT monitoring there (second last paragraph in Section 4). Fortunately through our lately funded Copernicus project ML-PhyTAO we are now able to investigate this issue further: https://marine.copernicus.eu/about/research-development-projects/2022-2024/ml-phytao

9. Line 269: First use of the acronym OMI, but it is not defined in the manuscript.

   We have added the full name 'Ocean Monitoring Indicator' in the revised manuscript.

10. Table 1: This is likely planned to be updated upon acceptance, but currently the documentation for the third row, concerning the new in situ PFT data states only "data to be submitted to PANGEA" and adds a citation that is not in the reference list. Considering that PANGEA offers the possibility of data publication with moratorium specifically to address situations of peer-review, I recommend that the data submission process, which might take time, be initiated and the information on the manuscript updated to reflect final citations/references and links. This point is also applicable to the data availability statement in lines 118-119 and 298-299.

    We agree with the reviewer to get a valid reference link of the data submitted to PANGAEA. We have already submitted it and will update in the manuscript the doi link once it is there. As PANGAEA is managed by the AWI, it is relatively quick to get the data set reviewed and available internally during the peer review stage.

11. Figure 1: The reference data set is presented in the Y-axis in subfigures (c) and (d). While this is unconventional, it is not per se an issue, however some statistics (regression slope, $R^2$ and MARD) are not symmetrical and do depend on which variable is taken as reference. The exception is model II regression for slope when using major axis estimation (cf. Legendre and Legendre, 2012), but that is not indicated.

Type II regression is used considering that both variables carry errors. We have now indicated it and defined all statistical terms involved in this study by adding section 2.5 in the revised manuscript.

12. Figure 3: The plots present values of "MDPD" though the acronym is not defined in the main text or figure caption. I note that in the main text MARD is defined, so even if it is a typo, the acronym for percentage still need to be formally defined.

As responded above, subsection 2.5 has been is added to define all statistical terms used in the study.

**2.5 Statistical metrics**
To evaluate the correction ensemble performance, relative difference (RD), median absolute difference (MAD) and median absolute relative difference (MARD) have been calculated based on the Chl$a$ data of each PFT, which are defined as below.
$RD_i$ = (Chl$a_i^{OLCI}$ - Chl$a_i^{Merged}$) / Chl$a_i^{Merged}$, where $i$ is the $i$th PFT

$$RD_{PFT} = \frac{(Chla\_PFT_{OLCI} - Chla\_PFT_{merged})}{Chla\_PFT_{merged}} * 100\% \qquad (1)$$

$$MAD_{PFT} = \text{median of } (Chla\_PFT_{OLCI} - Chla\_PFT_{merged}) \qquad (2)$$

$$MARD_{PFT} = \text{median of } \frac{|Chla\_PFT_{OLCI} - Chla\_PFT_{merged}|}{Chla\_PFT_{merged}} * 100\% \qquad (3)$$

To validate the corrected PFT Chl$a$ data with in situ data, statistical metrics including regression slope, determination coefficient ($R^2$), root mean square difference (RMSD, mg m$^{-3}$), and median percent difference (MDPD, %) have been used. For definition equations of these terms please refer to Xi et al. (2020). Note that the slope and $R^2$ are calculated in the base 10 logarithmic scale.

13. Figure 4: The indication of the regions in the map is not very legible. Though I believe there is no ambiguity where the regions are and what they cover, I think the visual presentation can be improved. The authors could consider using semitransparent filled boxes with a colour as in the legend by the side of the figure (removing the diagonal lines and names, and the grey shading of the polar areas).

We will consider the reviewer's comment and improve the map quality with indicated regions.

**Suggestions**

1. Chlorophyll $a$ is presented both with italicized (abstract) and roman (main body) "a". I suggest using the italicized form, as that is the form used in reference documents in the field of photosynthetic pigment analysis (e.g., Roy et al., 2011).

We have corrected all and used the italicized form, chlorophyll $a$ (Chl$a$), in the revised manuscript.

2. Communities vs Assemblages vs Ensembles vs Guild. Some terms in ecological literature have been used so loosely over the years such as to lose meaning. Currently, my opinion is to follow Fauth et al. (1996), who provides a classification and definition based on an intersection of three major areas: Phylogeny, Geography, and Resources. In their nomenclature, an ensemble is a "phylogenetically bounded group of species that use a similar set of resources within a community". Alternatively, assemblages are "phylogenetically related groups within a community". And "local guild" (though in this case, "local" seems inappropriate) is a group of "species that share a common resource and occur

in the same community". While not a perfect system, at least it is a structured one. Admittedly, I have used "phytoplankton assemblage" in my own work, though "ensemble" in their nomenclature or "guild" (considering phylogeny in algae is a loose link, and that phytoplankton includes cyanos) seem superior descriptors. I encourage the authors to consider this nomenclature system.

Thanks for the comment and the reference. We admit that we used phytoplankton community (composition) throughout the text as it is thought to be the most well accepted term used within the community (e.g., Cetinic et al. 2024) though it might not be ecologically rigorous. We have already used the term 'ensemble' for the machine learning approach, and "guild" is rather rarely seen in the literature though has been used in a few biological studies related to phytoplankton sizes (e.g. Sabbeta et al. 2005; Laraib et al. 2024).

3. "Composition structure" appears several times in the text. Consider using just "composition". It has been revised.

4. Line 33: The start of the phrase, "Dedicated to marine biogeochemistry, ", is unclear to me and seems unnecessary. Maybe just use a connector like "Therefore,"? It has been revised.

5. Line 39: Chla was already defined in line 28. It has been revised.

6. Line 50: "correction skill" -> "correction procedure". It has been revised.

7. Line 52: The corrected PFT time series" -> "The harmonized PFT time series". It has been revised.

8. Line 54: "For a longer goal" -> "Considering that ocean colour missions are planned to be continued into the next decade and beyond, ". It has been revised.

9. Line 59: "The PFT datasets with per-pixel uncertainty (product ref. no. 1 in Table 1) have been generated by the EOF-PFT approach adapted based on the version proposed by Xi et al. (2021). The updated algorithms (...)" -> "The PFT datasets with per-pixel uncertainty (product ref. no. 1 in Table 1) are produced with a modified version of the EOF-PFT approach proposed by Xi et al. (2021). The modified algorithms (...)". It has been revised.

10. Line 135: "cannot be assured" -> "could not be assured"? It has been revised.

11. Line 151: "The slope of the corrected dataset is close to one" -> "The slope of the regression when using the corrected dataset is close to one". It has been revised.

12. Line 289: I'm not sure if "white ocean" and "green ocean" are well established jargon, but if the jargon is not necessary to facilitate communication, I suggest describing the intended concepts directly.
They were defined by the Copernicus Marine Service and have been widely spread in the community: https://marine.copernicus.eu/services/monitoring-ocean. We added in the brackets after "white ocean" and "green ocean" their brief definations.

13. Equation 1: As it is, it seems to me a bit lost in the manuscript and it could be generalized. Maybe include after the call to it in line 143, "relative difference (RD in %) between the OLCI-derived and merged sensor-derived diatom Chla. RDs were calculated as: $RD_i = (Chla_i^{OLCI} - Chla_i^{Merged}) / Chla_i^{Merged}$, where $i$ is the $i$th PFT."

We modified and moved the equation to section 2.5.

14. All figures that include a global map: For visual consistency, I suggest that all global maps are presented in the same projection. I recommend the projection of Figure 6 to also be used in Figures 1, 2 and 5, or the Mollweide projection (equal area) to be use for all global maps.

We have applied the same projection for all other figures as in Figure 6 (Robinson projection in this case) in the revised manuscript.

15. Figure 3: A recommendation is to not extend the regression lines beyond the range of the X-axis data, as this is beyond the domain of the data used to calculate the regression.

We kept the regression line always in the same range as the X-axis limit for a better visual effect.

**References**

Fauth, J. E.; Bernardo, J.; Camara, M.; Resetarits, W. J.; Van Buskirk, J.; McCollum, S. A. 1996. Simplifying the Jargon of Community Ecology: A Conceptual Approach. The American Naturalist, 147(2), 282–286. doi: http://www.jstor.org/stable/2463205

Ho, Tin Kam. 1995. Random Decision Forests. Proceedings of the 3rd International Conference on Document Analysis and Recognition, Montreal, QC, 14–16 August 1995. pp. 278–282.

Legendre, P.; Legendre, L. 2012. Numerical Ecology. 3rd edition. Elsevier, Amsterdam.

Meyer, H.; Reudenbach, C.; Hengl, T.; Katurji, M.; Nauss, T. 2018. Improving performance of spatio-temporal machine learning models using forward feature selection and target-oriented validation. Environmental Modelling & Software 101, 1-9. doi: https://doi.org/10.1016/j.envsoft.2017.12.001

Roy, S.; Llewellyn, C. A.; Egeland, E. S.; Johnsen, G. (editors). Phytoplankton Pigments: Characterization, Chemotaxonomy, and Applications in Oceanography. Cambridge University Press, Cambridge, UK, 2011.

Xi, H.; Bretagnon, M.; Losa, S. N.; Brotas, V.; Gomes, M.; Peeken, I.; Alvarado, L. M. A.; Mangin, A.; Bracher, A. 2023. Satellite monitoring of surface phytoplankton functional types in the Atlantic Ocean over 20 years (2002-2021). Contribution to the 7th edition of Copernicus Marine Service Ocean State Report. State of the Planet 1-osr7, 5. doi: https://doi.org/10.5194/sp-1-osr7-5-2023, 2023.

**References**

Brando, V., Santoleri, R., Colella, S., Volpe, G., Di Cicco, A., Sammartino, M., González Vilas, L., Lapucci, C., Böhm, E., Zoffoli, M., Cesarini, C., Forneris, V., La Padula, F., Mangin, A., Jutard, Q., Bretagnon, M., Bryère, P., Demaria, J., Calton, B., Netting, J., Sathyendranath, S., D'Alimonte, D., Kajiyama, T., Van Der Zande, D., Vanhellemont, Q., Stelzer, K., Böttcher, M., Lebreton, C., 2024. Overview of Operational Global and Regional Ocean Colour Essential Ocean Variables Within the Copernicus Marine Service. Remote Sensing 16, 4588. https://doi.org/10.3390/rs16234588

Cetinić, I., Rousseaux, C., Carroll, I., Chase, A.P., Kramer, S.K. Werdell, J. et al. 2024. Phytoplankton composition from sPACE: Requirements, opportunities, and challenges. Remote Sensing of Environment, 302, 113964. https://doi.org/10.1016/j.rse.2023.113964.

Garnesson, P., Mangin, A., Bretagnon, M., and Jutard, Q.: EU Copernicus Marine Service Quality Information Document (QUID) for OC TAC Products OCEANCOLOUR OBSERVATIONS GlobColour, Issue 5.0, Mercator Ocean International, https://documentation.marine.copernicus.eu/QUID/CMEMS-OC-QUID-009-101to104-111-113-116-118.pdf (last access: 22 Aug 2024), 2024.

Laraib, M., Titocci, J., Giannakourou, A., Reizopoulou, S., Basset, A., 2024. Role of rare species on phytoplankton size–abundance relationships and size structure across different biogeographical areas. Diversity, 16, 98. https://doi.org/10.3390/d16020098

Li, Z., Sun, D., Wang, S., Huan, Y., Zhang, H., Liu, J., He, Y., 2023. A global satellite observation of phytoplankton taxonomic groups over the past two decades. Global Change Biology 29, 4511–4529. https://doi.org/10.1111/gcb.16766

Meyer, H., Reudenbach, C., Hengl, T., Katurji, M., Nauss, T., 2018. Improving performance of spatio-temporal machine learning models using forward feature selection and target-oriented validation. Environmental Modelling & Software 101, 1–9. https://doi.org/10.1016/j.envsoft.2017.12.001

Sabetta, L.A., Fiocca, L., Margheriti, F., Vignes, A., Basset,O.. and Ianni.C., 2005. Body size abundance distributions of nano- micro phytoplankton guilds in coastal marine ecosystems. Estuarine, coastal and shelf science. 63, 645-663. https://doi.org/10.1016/j.ecss.2005.01.009

Stock, A., Gregr, E.J., Chan, K.M.A., 2023. Data leakage jeopardizes ecological applications of machine learning. Nat Ecol Evol 7, 1743–1745. https://doi.org/10.1038/s41559-023-02162-1

Xi, H., Losa, S. N., Mangin, A., Soppa, M. A., Garnesson, P., Demaria, J., Liu, Y., d'Andon, O. H. F., and Bracher, A., 2020. A global retrieval algorithm of phytoplankton functional types: Towards the applications to CMEMS GlobColour merged products and OLCI data. Remote Sensing of Environment, 240, 111704, https://doi.org/10.1016/j.rse.2020.111704, 2020.

Xi, H., Losa, S. N., Mangin, A., Garnesson, P., Bretagnon, M., Demaria, J., Soppa, M. A., d'Andon, O. H. F., and Bracher, A., 2021. Global chlorophyll a concentrations of phytoplankton functional types with detailed uncertainty assessment using multi-sensor ocean color and sea surface temperature satellite products. Journal of Geophysical Research-Oceans, 126(5), https://doi.org/10.1029/2020JC017127, 2021.

Xi, H., Bretagnon, M., Losa, S.N., Brotas, V., Gomes, M., Peeken, I., Alvarado, L.M.A., Mangin, A., and Bracher, A., 2023a. Satellite monitoring of surface phytoplankton functional types in the Atlantic Ocean over 20 years (2002-2021). 7th Copernicus Ocean State Report. State of the Planet, 1-osr7, 5. https://doi.org/10.5194/sp-1-osr7-5-2023

Xi, H., Peeken, I., Gomes, M., Brotas, V., Tilstone, G., Brewin, R. J. W., Dall'Olmo, G., Tracana, A., Alvarado, L. M. A., Murawski, S., Wiegmann, S., and Bracher, A., 2023b. Phytoplankton pigment concentrations and phytoplankton groups measured on water samples collected from various expeditions in the Atlantic Ocean from 71°S to 84°N, PANGAEA [data set], https://doi.org/10.1594/PANGAEA.954738.

Xi, H., Peeken, I., Nöthig, E.M., Kraberg, A., Metfies, K., Bretagnon, M., Mehdipour, E., Lampe, V., Mangin, A., and Bracher, A., 2024. Is the surface phytoplankton community composition changing in the Fram Strait? Ocean Optics Conference XXVI, Las Palmas Spain, 6-11 October 2024.

Zhang, Y., Shen, F., Li, R., Li, M., Li, Z., Chen, S., Sun, X., 2024. AIGD-PFT: the first AI-driven global daily gap-free 4 km phytoplankton functional type data product from 1998 to 2023. Earth Syst. Sci. Data 16, 4793–4816. https://doi.org/10.5194/essd-16-4793-2024

Zoffoli, M.L., Brando, V., Volpe, G., González Vilas, L., Davies, B.F.R., Frouin, R., Pitarch, J., Oiry, S., Tan, J., Colella, S., Marchese, C., 2025. CIAO: A machine-learning algorithm for mapping Arctic Ocean Chlorophyll-a from space. Science of Remote Sensing, 11, 100212. https://doi.org/10.1016/j.srs.2025.100212

---

## Author Comment (AC2)

**Consistent long-term observations of surface phytoplankton functional types from space**

Hongyan Xi[1*,] Marine Bretagnon[2], Ehsan Mehdipour[1,3], Julien Demaria[2], Antoine Mangin[2], Astrid Bracher[1,4]

[1]Alfred Wegener Institute, Helmholtz-Centre for Polar and Marine Research, Bremerhaven, 27570, Germany
[2]ACRI-ST, Sophia Antipolis Cedex, France
[3]School of Business, Social & Decision Sciences, Constructor University, Bremen, Germany
[4]Institute of Environmental Physics, University of Bremen, Bremen, 28359, Germany

*Correspondence to*: Hongyan Xi (hongyan.xi@awi.de)

**Author Comments in response to Referee #2**

This paper introduces a machine learning-based correction method to harmonize global phytoplankton functional type (PFT) data obtained from various ocean color sensors, addressing the discrepancies caused by differences in sensor characteristics. The authors propose the use of a random forest-based ensemble learning method (MLBE) for this task. By correcting the OLCI-derived PFT data to match the merged sensor-derived PFT data, this method ensures more consistent and reliable global PFT observations. The study demonstrates the utility of this correction for analyzing long-term trends in PFTs, revealing significant changes in the biomass of diatoms and dinoflagellates, while showing more stable trends for haptophytes and prokaryotes. Additionally, it examines anomalies in PFTs, noting significant increases in diatom and dinoflagellate Chla concentrations, particularly in higher latitudes and coastal regions.

Overall, the paper is clear in its objectives, and the methodologies employed are robust. The innovative use of machine learning to calibrate and harmonize PFT data from different sensors is a particularly valuable contribution. This approach significantly improves the accuracy and reliability of the resulting PFT time series, showcasing the potential of machine learning in enhancing ocean color application. Given the importance and novelty of this research, I recommend the publication of this paper. Before final acceptance, I have a few suggestions (listed below) that I hope the authors will consider to further refine and enhance the quality of the work.

We thank very much the reviewer for the positive feedback and constructive comments. We have carefully considered the suggestions during the revision. Below please find our response/clarificafication to each comment.

1. During model training, the authors randomly partitioned the dataset into a training set (70%) and 100 test sets (30%), achieving good validation results. However, the inclusion of latitude and longitude as input features may raise concerns about potential data leakage and shortcut learning, as spatial dependencies could result in overly optimistic estimates of model accuracy (https://doi.org/10.1038/s41559-023-02162-1). To mitigate this risk and improve the rigor of the validation, I suggest the authors incorporate temporal partitioning in addition to random partitioning. By dividing the dataset based on time and ensuring that the training and test sets are

strictly independent in terms of temporal coverage, the MLBE model's ability to generalize across both time and space can be more rigorously assessed.

We appreciate the reviewer's suggestion and the reference provided. We understand the logic and reason why the inclusion of temporal partitioning is important. However, we don't think it applies to our case here. Our MLBE model is basically a correction scheme, that is trained based on 12 months satellite data spaning only one year (the overlapping period of the two sensor sets), with the model we wanted to set up a regression model through random forest learning trying to identify better the spatial variation of the PFT data from the two sensor sets, so that it can fit one pattern to the other on the whole global scale. We considered all pixel data (over 4 million available data points) from the 12 monthly products, and we wanted to cover as complete as possible the whole global region to make sure the training learns the pattern globally. However, by applying the suggested temporal partitioning we would lose data, e.g., in high latitudes, if we exclude a certain month in the training. This may cause biases in the learning process. Then the trained model would very likely not be applicable to the test data set (because though they would be temporally independent from the training set, the spatial information in e.g. high latitudes which is not included in the training set could not be learnt from the training and thus the ML model might fail in the test data and also in the global products in later years).

Though we applied a straightforward random splitting in this study, the training and test data sets were nearly homogeneously divided over space and time by the random splitting due to the large amount of the data points (> 4 millions), as shown by (CDF, for each of the input variables) in Fig. R1 the cumulative distribution function. This ensures that the trained model takes the most of the knowledge of the available data sets within the limited time period that can be used in the correction model.

[Figure]

Figure R1. Cumulative distribution functions of input variables (PFT, lat, lon) involved in the MBLE training set, taking diatom as representative.

Reviewer 1 also posted a similar comment and we understand that there are limitations existing in the training and testing procedure and a discussion is necessary to clarify this point. Therefore, we add a paragraph of discussion about model caveats to cover this aspect:

"However, the MLBE model training was based on 12 months satellite data spanning only one year (the overlapping period of the two sensor sets), trying to identify the spatial variation of the PFT data from the two sensor sets, so that it could fit one pattern to the other on the whole global scale. It has been reported that random splitting between training and test sets may produce data leakages (Meyer et al., 2018; Stock et al., 2023) which result in overoptimistic performance in the test data but less good performance in actual applications to other data sets. To avoid data leakage data

temporal partitioning has been suggested to ensure that the training and data sets are independent. However, random split was applied in the study as the temporal partitioning does not apply to our case. The MLBE model is basically a correction scheme trained based on all pixel data (over 50 million available data points) from 12 monthly PFT products. The purpose was to cover as complete as possible the global region to ensure that the training learns the pattern globally. By applying the suggested temporal partitioning we would lose data, e.g., in high latitudes, if we exclude a certain month in the training. This can cause biases in the learning process, then the trained model would very likely not be applicable to either the test set or other data sets that contain the missing periods. The straightforward random splitting in our study ensured the homogeneous splitting between the training and test data sets over space and time thanks to the large amount of data points, so that the trained model learned the most knowledge from the available data within the limited time period. Though such random partitioning has been widely used (e.g., Li. et al. 2023; Zoffoli, et al. 2025), one should keep in mind that having data for only a single year is challenging because the year may present conditions that are specific to that year only which may cause unrealistic predictions for other years. It is therefore noteworthy that target-oriented data splitting and cross-validation such as considering spatial and temporal blocks should be applied in machine learning based studies when data set allows (e.g., Zhang et al. 2024)."

2. When calculating the relative difference (RD in %) using PFT data, is the RD calculation performed on the log-transformed data or the raw PFT data? It is recommended to clarify this in the paper.

The RD calculation is based on non-log transformed (real) PFT Chla concentrations to describe the true relative difference between products from two different sensors, similar to the other commonly used statistical parameters in ocean color models such as mean percentage difference, which are also based on the Chla concentration (e.g. Xi et al. 2020; 2021). This has been clarified in the revised manuscript. We have added a subsection to clarify it better:

**2.5 Statistical metrics**
To evaluate the correction ensemble performance, relative difference (RD), median absolute difference (MAD) and median absolute relative difference (MARD) have been calculated based on the Chl$a$ data of each PFT, which are defined as below.
$RD_i = (Chla_i^{OLCI} - Chla_i^{Merged}) / Chla_i^{Merged}$, where $i$ is the $i$th PFT

$$RD_{PFT} = \frac{(Chla\_PFT_{OLCI} - Chla\_PFT_{merged})}{Chla\_PFT_{merged}} * 100\% \qquad (1)$$

$$MAD_{PFT} = median\ of\ (Chla\_PFT_{OLCI} - Chla\_PFT_{merged}) \qquad (2)$$

$$MARD_{PFT} = median\ of\ \frac{|Chla\_PFT_{OLCI} - Chla\_PFT_{merged}|}{Chla\_PFT_{merged}} * 100\% \qquad (3)$$

To validate the corrected PFT Chl$a$ data with in situ data, statistical metrics including regression slope, determination coefficient ($R^2$), root mean square difference (RMSD, mg m$^{-3}$), and median percent difference (MDPD, %) have been used. For definition equations of these terms please refer to Xi et al. (2020). Note that the slope and $R^2$ are calculated in the base 10 logarithmic scale.

3. Similarly, when calculating the PFT anomaly, is the calculation performed on the log-transformed data or the raw PFT data?

   The PFT relative anomaly (%) was also calculated based on the original PFT Chl-a values. This has been clarified in the revised manuscript in section 2.5.

4. When calculating the average, it is important to clarify whether the authors computed a weighted average based on factors such as latitude, or if they simply calculated the unweighted mean of all pixels. Given that PFT variations are primarily observed in high-latitude regions, using a latitude-weighted average would be more reasonable.

   We agree with the two reviewers about the area distortion when calculating the mean spatially, and that a latitudinal weighted average can compensate the geographical distortion by taking into account proportional contribution. We therefore have modifed the calculations of the global and the four regional averages when generating PFT time series. The latitude-weighted averaging was applied to the logarithmic transformed Chla concentrations to get the log based mean which are then converted to their natural values. So for each monthly product over a certain region, the average was calculated based on the equation below:

   $$Mean_{chla\_diatom} = \exp\left(\frac{\sum \cos(lat) \cdot \ln(Chla_{diatom})}{\sum \cos(lat)}\right).$$

   We have added the following text in Section 2.4 of the revised manuscript: "PFT time series of different spatial scales were calculated by applying the weighted average (taking cosine of the latitude as weights) to the monthly PFT data over the defined regions, to take into account the proportional contribution of each pixel to the global surface ocean due to area distortion in the gridded dataset. The latitude-weighted averaging was applied to the logarithmic transformed PFT Chla to get the log-based mean which are then converted to natural values."

   The time series plots in Figure 4 have also been updated, showing overall slight changes in the trends, however the Chla magnitudes of the PFT time series at global scale are in general slightly lower for the weighted average, except for prokaryotes. This is mainly due to much lower prokaryotes Chla concentrations in high latitudes (compared to lower latitudes) contributed less with the latitude weighting applied, contrasting to other PFTs which have in general higher Chla in higher latitudes. The trend of the global diatom Chla was slightly decreased (from 0.0014 to 0.0011 mg m-3 per decade), while their increase at high latitudes is still very prominent, as the proportional weights are also considered in the 'divider' which is the weighted total number of observations. Accordingly, we have updated Figure 4 and the statistical description related to this figure in Section 3 of the revised manuscript.

[Figure]

**Figure 4. Panel (a): Updated (corrected) time series of the five PFT Chla based on the global mean from 2002 to 2023. Merged products cover the period of July 2002-April 2017 (indicated with dots), and OLCI products are for May 2016-Dec 2023 (indicated with crosses). Note that the OLCI products have been corrected to merged products based on MLBE. Panel (b): Trends of diatoms, haptophytes, dinoflagellates, green algae and prokaryotes Chla on the global scale and four regional scales (the North Atlantic Ocean, the Mediterranean Sea, the Arctic Ocean and the Southern Ocean), respectively. Trend slopes per decade with uncertainties have been indicated with significant trends marked with an asterisk (*).**

5. The MLBE model demonstrates excellent performance, but it would be valuable to explore whether it can be applied to data from other sensors. Given the potential for

future expansion, I recommend that the authors include a brief discussion in the paper about potential future work, particularly how the correction method could be further improved or extended to incorporate data from other satellite sensors. This discussion would not only highlight the adaptability and scalability of the method to other satellite datasets but also significantly enhance the broader impact and relevance of this research. I believe that including this discussion would add considerable value to the paper.

We agree with the reviewer. Indeed, such discussion is very necessary. We have added the following text in the discussion. We would also like to point out that this manuscript as a contribution to the Ocean State Report needs comply with the length limit, hence we tried to include the discussion as concise as possible.

"The correction scheme proposed in this study is specifically designed to address inter-sensor data inconsistencies in the current Copernicus Marine Service PFT products. The present trained model can be only used to correct the OLCI-derived PFT product to match the merged sensor-derived product. However, the underlying technical framework is adaptable to other common ocean color products, such as optical properties derived from multiple sensors, thereby enhancing the overall continuity and consistency of ocean color data. As a rapidly emerging and powerful technique, machine learning can be further leveraged in ocean color data services, supporting agencies and data platforms in delivering high-quality, consistent operational products."

**References**

Li, Z., Sun, D., Wang, S., Huan, Y., Zhang, H., Liu, J., He, Y., 2023. A global satellite observation of phytoplankton taxonomic groups over the past two decades. Global Change Biology 29, 4511–4529. https://doi.org/10.1111/gcb.16766

Meyer, H., Reudenbach, C., Hengl, T., Katurji, M., Nauss, T., 2018. Improving performance of spatio-temporal machine learning models using forward feature selection and target-oriented validation. Environmental Modelling & Software 101, 1–9. https://doi.org/10.1016/j.envsoft.2017.12.001

Stock, A., Gregr, E.J., Chan, K.M.A., 2023. Data leakage jeopardizes ecological applications of machine learning. Nat Ecol Evol 7, 1743–1745. https://doi.org/10.1038/s41559-023-02162-1

Xi, H., Losa, S. N., Mangin, A., Soppa, M. A., Garnesson, P., Demaria, J., Liu, Y., d'Andon, O. H. F., and Bracher, A., 2020. A global retrieval algorithm of phytoplankton functional types: Towards the applications to CMEMS GlobColour merged products and OLCI data. Remote Sensing of Environment, 240, 111704, https://doi.org/10.1016/j.rse.2020.111704

Xi, H., Losa, S. N., Mangin, A., Garnesson, P., Bretagnon, M., Demaria, J., Soppa, M. A., d'Andon, O. H. F., and Bracher, A., 2021. Global chlorophyll a concentrations of phytoplankton functional types with detailed uncertainty assessment using multi-sensor ocean color and sea surface temperature satellite products. Journal of Geophysical Research-Oceans, 126(5), https://doi.org/10.1029/2020JC017127

Zhang, Y., Shen, F., Li, R., Li, M., Li, Z., Chen, S., Sun, X., 2024. AIGD-PFT: the first AI-driven global daily gap-free 4 km phytoplankton functional type data product from 1998 to 2023. Earth Syst. Sci. Data 16, 4793–4816. https://doi.org/10.5194/essd-16-4793-2024

Zoffoli, M.L., Brando, V., Volpe, G., González Vilas, L., Davies, B.F.R., Frouin, R., Pitarch, J., Oiry, S., Tan, J., Colella, S., Marchese, C., 2025. CIAO: A machine-learning algorithm for mapping Arctic Ocean

Chlorophyll-a from space. Science of Remote Sensing 11, 100212. https://doi.org/10.1016/j.srs.2025.100212